# Plasma membrane damage causes NLRP3 activation and pyroptosis during *Mycobacterium tuberculosis* infection

Kai S. Beckwith[1,4], Marianne S. Beckwith [1,4], Sindre Ullmann[1], Ragnhild S. Sætra [1], Haelin Kim[1], Anne Marstad[1], Signe E. Åsberg[1], Trine A. Strand[1], Markus Haug [1], Michael Niederweis [2], Harald A. Stenmark [1,3] & Trude H. Flo [1✉]

*Mycobacterium tuberculosis* is a global health problem in part as a result of extensive cytotoxicity caused by the infection. Here, we show how *M. tuberculosis* causes caspase-1/NLRP3/gasdermin D-mediated pyroptosis of human monocytes and macrophages. A type VII secretion system (ESX-1) mediated, contact-induced plasma membrane damage response occurs during phagocytosis of bacteria. Alternatively, this can occur from the cytosolic side of the plasma membrane after phagosomal rupture in infected macrophages. This damage causes $K^+$ efflux and activation of NLRP3-dependent IL-1β release and pyroptosis, facilitating the spread of bacteria to neighbouring cells. A dynamic interplay of pyroptosis with ESCRT-mediated plasma membrane repair also occurs. This dual plasma membrane damage seems to be a common mechanism for NLRP3 activators that function through lysosomal damage.

[1] Centre of Molecular Inflammation Research, Department of Clinical and Molecular Medicine, Norwegian University of Science and Technology (NTNU), 7491 Trondheim, Norway. [2] Department of Microbiology, The University of Alabama at Birmingham, Birmingham, AL 35294, USA. [3] Centre for Cancer Cell Reprogramming, Institute of Clinical Medicine, University of Oslo, Montebello, Oslo 0379, Norway. [4] These authors contributed equally: Kai S. Beckwith, Marianne S. Beckwith. ✉email: trude.flo@ntnu.no

**M**ycobacterium tuberculosis (Mtb) is a human pathogen, causing about 1.6 million deaths per year[1]. A pathological hallmark of Mtb infection is extensive necrosis in infected tissues[2]. Necrosis has long been regarded as an unregulated type of cell death, but recently several programmed necrotic pathways have been identified[3,4]. A highly inflammatory form of programmed necrosis is pyroptosis, occurring mainly in myeloid cells after pattern-recognition receptor activation. In the classical pathway, activation of nucleotide-binding oligomerisation domain-like receptors (NLRs) or absent in myeloma 2 (AIM2)-like receptors (ALRs) by pathogen- or self-ligands drives the assembly of an inflammasome consisting of oligomerised NLRs or ALRs, the adaptor apoptosis-associated speck-like protein containing a CARD (ASC) and caspase-1 (refs. [5–7]). Autocatalytic activation and cleavage of caspase-1 enables cleavage of pro-inflammatory cytokines interleukin (IL)-1β and IL-18, as well as the pore-forming molecule gasdermin D (GSDMD)[8,9]. IL-1β is released through GSDMD pores, and in larger amounts during pyroptosis, the lytic cell death that often follows GSDMD pore formation[10–13].

IL-1β is a critical host-protective cytokine during Mtb infection, and canonical NLRP3 (NOD-, LRR- and pyrin-domain containing protein 3) and AIM2 inflammasome activation have been implicated in IL-1β release during Mtb infection in mouse and human macrophages[14–16]. However, NLRP3-independent routes to IL-1β release have been reported in mouse infection models, leaving the role for NLRP3 in vivo less clear[17,18]. The agonist of AIM2 is double-stranded DNA[19–21], while the direct agonists of NLRP3 are not known. With few exceptions, two steps are required for NLRP3 activation: the priming signal involves increased expression of pro-IL-1β as well as inflammasome components such as NLRP3 itself, while the second signal is characterised by a range of cell damage events such as potassium ($K^+$) and chloride ($Cl^-$) efflux, mitochondrial dysfunction, metabolic changes, calcium fluxes, trans-Golgi disassembly and lysosomal damage[7,22–31]. $K^+$ efflux in particular is considered a key determinant of NLRP3 activation for a range of triggers[27], with some exceptions involving mitochondrial disruption and mitochondrial reactive oxygen species production, e.g. by the small-molecule compound Imiquimod[28]. Mitochondrial dysfunction seems closely tied to Mtb-induced necrosis as well[32,33], but whether this is related to inflammasome activation is not clear. Inflammasome activation and pyroptosis by Mtb are dependent on the type VII secretion system 6-kDa early secretory antigenic target (ESAT-6) secretion system 1, ESX-1. ESX-1 secretes a range of protein substrates, and is postulated to mediate phagosome permeabilisation with full or partial translocation of Mtb into the cytosol[34–38]. Phagosomal permeabilisation presumably allows the release of Mtb DNA and activation of AIM2 (ref. [39]), and release of active cathepsin B from damaged phagolysosomes has been proposed as a trigger for NLRP3 (refs. [23,40,41]). However, off-target inhibitor effects and negative results in knockout mouse experiments have left the question open whether cathepsin release plays a role in NLRP3 inflammasome activation[27,42–44].

Here, we use time-lapse- and correlative microscopy to spatiotemporally resolve Mtb-induced inflammasome activation and cell death at the single-cell level. Our findings identify a common mechanism where plasma membrane (PM) damage caused by Mtb or crystals triggers inflammasome activation, IL-1β release and pyroptosis, unless the membrane damage is balanced by repair mediated by the endosomal sorting complexes required for transport (ESCRT) machinery[45].

## Results

### Mtb induces canonical NLRP3 activation followed by pyroptosis.
Based on previously published findings[14–16,18,39], we hypothesised

that Mtb induces the assembly of one of the ASC-dependent inflammasomes, NLRP3 or AIM2, upon infection of macrophages. THP-1 macrophages expressing ASC-GFP[46] were infected with Mtb H37Rv constitutively expressing BFP (Mtb-BFP) and imaged after 24 h of infection or over an entire 24-h infection time course by time-lapse microscopy. The cell-impermeable DNA dye DRAQ7 was present in the medium to assess cell death. Mtb induced ASC-speck formation in infected macrophages (Fig. 1), and specks were exclusively localised to dead cells as indicated by DRAQ7-positive nuclei (Supplementary Fig. 1a). The number of dead cells, ASC specks and IL-1β secretion increased with increasing multiplicity of infection (MOI, Supplementary Fig. 1b). Treatment with the specific NLRP3 inhibitor MCC950 (ref. [47]) or increased extracellular concentrations of KCl inhibited the formation of ASC specks, demonstrating that Mtb infection results in potassium efflux-driven activation of the NLRP3 inflammasome (Fig. 1a). Correspondingly, IL-1β secretion was inhibited by MCC950 and KCl, as well as the caspase inhibitors Z-VAD-FMK (pan-caspase) or VX765 (caspase-1). Treatment with Z-VAD, VX765, MCC950 or KCl all reduced cell death measured by DRAQ7 influx after 24 h, and DRAQ7 imaging results corresponded well to lactate dehydrogenase (LDH) cytotoxicity measurements in THP-1 cells (Supplementary Fig. 1c). Knocking down NLRP3 using CRISPR–Cas9 completely inhibited ASC-speck formation and reduced Mtb-induced cell death in a corresponding fraction of cells (Fig. 1b; Supplementary Fig. 1d). Similar results were observed with inhibitors and NLRP3 knockdown using the canonical NLRP3 activator LPS and nigericin (Supplementary Fig. 1e, f). Live-cell time-lapse microscopy of single Mtb-infected cells revealed nuclear DRAQ7 within few minutes after ASC-speck formation in pyroptotic cells, while DRAQ7 influx was absent for a period after ASC-speck formation in THP-1 cells deficient in GSDMD (Fig. 1c; Supplementary Fig. 1d), implicating cell lysis by GSDMD pores downstream of NLRP3 inflammasome activation by Mtb.

To address the physiological relevance of these findings, we measured cell death and IL-1β release in primary human monocytes and macrophages following 24-h infection with Mtb-BFP (Fig. 1d, e). Treatment with MCC950 or KCl abolished IL-1β release and significantly reduced cell death, implicating NLRP3-mediated inflammasome activation and pyroptosis also in primary human monocytes and macrophages. Caspase inhibitors were similarly effective in inhibiting IL-1β release, but less effective in inhibiting cell death in primary cells, as previously reported[48,49].

Together, our data demonstrate that Mtb induces potassium efflux-driven activation of the canonical NLRP3 inflammasome upon infection of human monocytes and macrophages, followed by GSDMD-dependent pyroptotic cell death with release of IL-1β.

To gain a better understanding of the kinetics and prevalence of pyroptosis caused by Mtb, we imaged THP1-ASC-GFP cells infected with Mtb-BFP for 40 h by time-lapse microscopy. We noted the cumulative number of dead cells by pyroptosis, apoptosis and necrosis, scored as described in "Methods" and in Fig. 1f and the corresponding Supplementary Movies 1, 2 and 3. Pyroptosis was the dominating form of cell death during the first 24 h of infection. In line with this result, the necroptosis inhibitors Nec1s (receptor-interacting protein kinase 1 (RIPK1) inhibitor) and GSK'872 (RIPK3 inhibitor) did not affect cell death or ASC-speck formation in THP-1 cells (Supplementary Fig. 1g). However, in primary human monocytes and macrophages, there was also a contribution of RIPK3-mediated necroptosis 24 h post infection (Supplementary Fig. 1h, i). From time-lapse imaging, we observed that pyroptosis and necrosis primarily occurred in infected cells, while the proportion of apoptotic cell death was larger in uninfected bystander cells (Fig. 1f). In addition, the

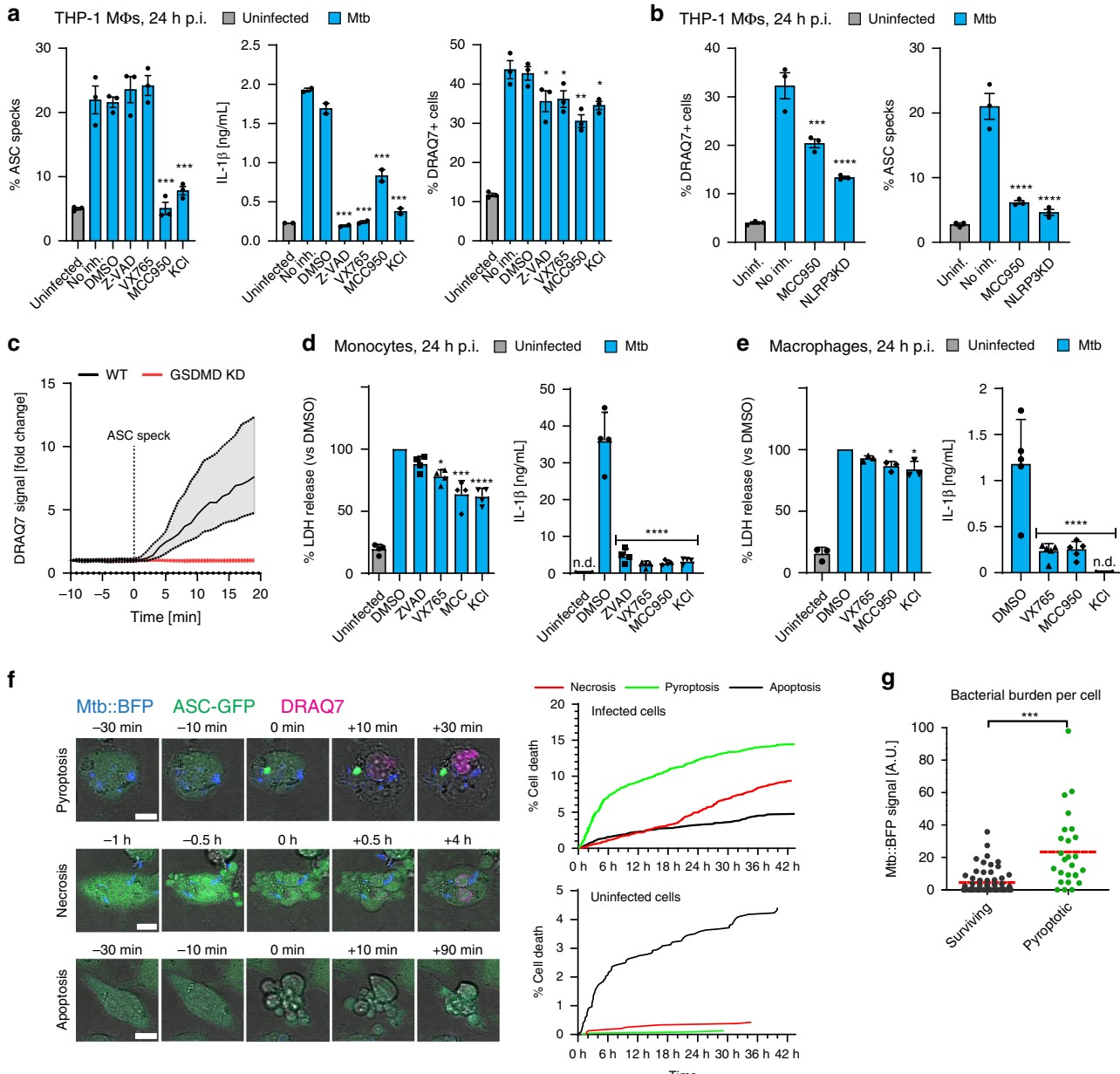

**Fig. 1 Mtb H37Rv infection induces canonical NLRP3 activation.** THP-1 ASC-GFP macrophages were infected by Mtb-BFP and imaged by time-lapse microscopy or at 24 h post infection (p.i.). **a** Cells were treated with DMSO control, caspase inhibitors (Z-VAD-FMK (pan-caspase) or VX765 (caspase-1)) or NLRP3 inhibitors (MCC950, KCl). DRAQ7+ cells and ASC specks were quantified at 24 h p.i. IL-1β release was determined by ELISA. **b** THP-1 ASC-mNeonGreen (WT) or NLRP3 KD cells were infected by Mtb and imaged 24 h p.i. **c** To assess the role of GSDMD in cell death in cells with ASC speck formation, THP-1 ASC-mNeonGreen (WT) or GSDMD KD cells were infected with Mtb and imaged live. DRAQ7 intensity during ASC-speck formation was measured in single cells in WT (n = 57) and GSDMD KD (n = 81) cells. Median ± IQR are shown. **d** Primary human monocytes were treated with inhibitors and infected with Mtb, and LDH and IL-1β release were determined 24 h p.i. n.d.—below detection limit. n = 4 biologically independent samples examined over two independent experiments. **e** Primary human macrophages were treated with inhibitors and infected with Mtb. LDH and IL-1β release were determined 24 h p.i. n = 3 and n = 5 biologically independent samples for LDH and IL-1β data, respectively. **f** The prevalence of pyroptosis was assessed by quantifying the cumulative number of pyroptotic, necrotic and apoptotic cell death events in THP-1 ASC-GFP cells from a 45-h time-lapse experiment during infection with Mtb. Representative cell death events are shown. Scale bars 10 μm. Data representative of two independent experiments. **g** The bacterial burden (intracellular Mtb-BFP fluorescence) was determined immediately before ASC-speck formation in pyroptotic cells (n = 26), and at the average time of ASC-speck formation in cells surviving the 24-h experiment (n = 61). Lines indicate median values. Bars show mean ± s.e.m. in (**a**, **b**) and mean ± s.d. in (**d**, **e**). *p < 0.05, **p < 0.01, ***p < 0.001, ****p < 0.0001, by one-way ANOVA with two-sided Dunnett's test in (**a**, **b**), repeated measures ANOVA with two-sided Dunnett's test in (**d**, **e**) and two-sided Mann–Whitney test in (**g**). Data in (**a–c**) representative of three independent experiments with n > 2000 cells in technical triplicates per condition. Source data are provided as a Source Data file.

bacterial burden was higher in cells that formed ASC specks than in infected cells that did not (Fig. 1g). These results indicate that pyroptosis from Mtb infection is a cell-intrinsic effect, i.e. a direct result of infection within single cells.

We reasoned the relative reduction in pyroptosis, and the increase in necrosis 24–48 h of infection could be due to prolonged stimulation of Toll-like receptor (TLR) signalling, which has been reported to inhibit NLRP3 inflammasome activation[50]. Indeed, we observed that prolonged treatment (24 h) of THP-1 cells with TLR2 or TLR4 ligands prior to infection with Mtb decreased cell death and ASC speck formation to a similar extent as treatment with MCC950 (Supplementary Fig. 1j).

**Pyroptosis causes severe damage and allows for spread of Mtb.** Cell death is accompanied with morphological changes that can inform about the processes involved in execution of a particular cell death pathway. We investigated the ultrastructural features of pyroptosis using focused ion beam–scanning electron microscopy (FIB–SEM), allowing near-isotropic 3D resolution. Cells of interest were located by light microscopy using a correlation system we developed previously[51]. Uninfected THP1-ASC-GFP cells displayed a normal, healthy morphology (Fig. 2a). In comparison, infected, pyroptotic cells (i.e. with visible ASC specks) had severe ultrastructural disruptions, including fragmented and occasionally split nuclear membranes, large PM disruptions, organelle damage and leakage of content to the extracellular space (Fig. 2b). Mtb displayed a largely intact morphology, and some bacteria were observed in contact with the extracellular space, suggesting a means of escape from pyroptotic cells. Cells that were treated with LPS and nigericin and formed ASC specks displayed similar ultrastructural features as Mtb-infected pyroptotic cells, with nuclear and PM disruptions, cytosol leakage and organelle damage (Supplementary Fig. 2a).

To distinguish morphological changes that occurred as a consequence of pyroptosis from those that occurred prior to or during inflammasome activation, we infected cells for 5 h with Mtb-BFP in the presence of the caspase inhibitors Z-VAD and VX765. Cells were imaged live during infection with DRAQ7 in the medium, and chemically fixed within 2 min of the last captured frame. Cells with an ASC speck but without influx of DRAQ7 and no visible changes to cell morphology were chosen for further imaging by FIB–SEM. The morphology was similar to uninfected cells, with intact organelles, a continuous PM and a double nuclear membrane (Fig. 2c). Similar results were observed in THP-1 GSDMD KD cells (Supplementary Fig. 2b). Hence, the dramatic morphological changes in pyroptotic cells are a consequence of the pyroptotic cell death process itself.

Next, we investigated the structure of the ASC speck using the correlative imaging approach. ASC specks were visible in SEM images as aggregates with a branched structure and a size of about 1–2.5 μm in THP1-ASC-GFP cells, both when infected with Mtb and when treated with LPS and nigericin (Fig. 2d, e). The structure corresponds well to that previously observed in a zebrafish model[52]. The speck in the living (Z-VAD + VX765-treated) cell was located in an open space in the cytosol of the cell, seemingly devoid of other organelles. To ensure that the ASC-speck structure was not a consequence of ASC overexpression or tagging, we treated primary human monocytes with LPS and nigericin, and visualised active caspase-1 by FAM-YVAD-FMK (FLICA). Caspase-1 is stabilised in its active form on the inflammasome[53], and we therefore hypothesised that the point with the highest fluorescence intensity from FLICA treatment would be the point where the speck was assembled. Indeed, by correlative light and EM (CLEM), we identified a similar

branched structure in LPS and nigericin-treated primary human monocytes, albeit with a smaller size than that observed in THP-1 cells (~0.8 μm). Our data demonstrate that ASC specks also form in primary cells, without tagging or overexpression of selected proteins.

To elaborate on the possible consequences of pyroptosis on Mtb spreading, we investigated pyroptosis during time-lapse imaging. On average, ~25% of Mtb bacilli were lost into the medium immediately after pyroptosis (Fig. 2f), while the rest were trapped in the pyroptotic cell. However, Mtb that was trapped in pyroptotic cells was phagocytosed by neighbouring cells. Further, this induced ASC-speck formation with pyroptosis in the new host cell, thus propagating inflammasome activation and cell death (Fig. 2g; Supplementary Movie 4). This indicates that pyroptosis enables some bacteria to spread immediately after cell death, and that bacteria that are trapped in the ghost cell can also contribute to further local spread of infection by efferocytosis.

**Mitochondrial depolarisation accompanies pyroptosis.** It has been reported that Mtb lacking the region of difference 1 (MtbΔRD1), where most ESX-1-related genes are located, is deficient in inflammasome activation[14,15,18]. Our results are in line with these reports, as MtbΔRD1 caused significantly less ASC-speck formation and cell death of THP-1 cells than wild-type Mtb (Fig. 3a). We also observed that Mtb grown in the absence of the detergent tween-80 is more potent in inflammasome activation than Mtb grown in the presence of tween (our regular growth condition). Detergent is found by others to perturb the mycobacterial capsule, in particular decreasing the abundance of ESX-1-related proteins on the Mtb surface/capsule[54,55]. Our data thus show that inflammasome activation correlates with ESX-1 activity and the integrity of the Mtb capsule, suggesting that deposition of ESX-1-related factors on the Mtb surface/capsule is important for Mtb inflammasome activation.

Tuberculosis-necrotising toxin (TNT, the C-terminal end of the channel protein with necrosis-inducing toxin (CpnT)) is secreted from Mtb and released into the host cell cytosol in an ESX-1-dependent manner, leading to macrophage necroptosis[33,56,57]. We therefore wanted to investigate if TNT contributes to the ESX-1-dependent inflammasome activation. To this end, we infected THP-1-ASC-GFP macrophages with wild-type Mtb, MtbΔcpnT or MtbΔcpnT complemented with cpnT containing either the catalytically active or inactive form of TNT (MtbΔcpnT::cpnT and MtbΔcpnT::cpnT*, respectively). We found that all Mtb mutants and complemented strains were similarly competent in inflammasome activation, and that treatment with MCC950 had a similar effect across bacterial strains in inhibiting ASC speck formation (Fig. 3b; Supplementary Fig. 3a). This indicates that TNT does not play a role in Mtb-induced inflammasome activation in our system.

Finally, we assessed the possible involvement of mitochondrial membrane perturbation prior to inflammasome activation and pyroptosis, an event that could also be linked to the activity of ESX-1 and related proteins. Mitochondrial damage has been strongly implicated in inflammasome activation in general, and in Mtb-induced necrosis[24,28,32,33,58,59]. In particular, infection with Mtb H37Rv was reported to cause a loss in mitochondrial membrane potential ($\Delta\Psi_m$), which was inhibited by cyclosporine A (CsA), a cyclophilin D inhibitor that prevents formation of the mitochondrial permeability transition pore[33,60]. However, we did not see an effect of CsA on Mtb-induced inflammasome activation and pyroptosis after 24 h of infection (Fig. 3c). In addition, we monitored the $\Delta\Psi_m$ by live-cell imaging of cells infected with Mtb-BFP and labelled with tetramethylrhodamine

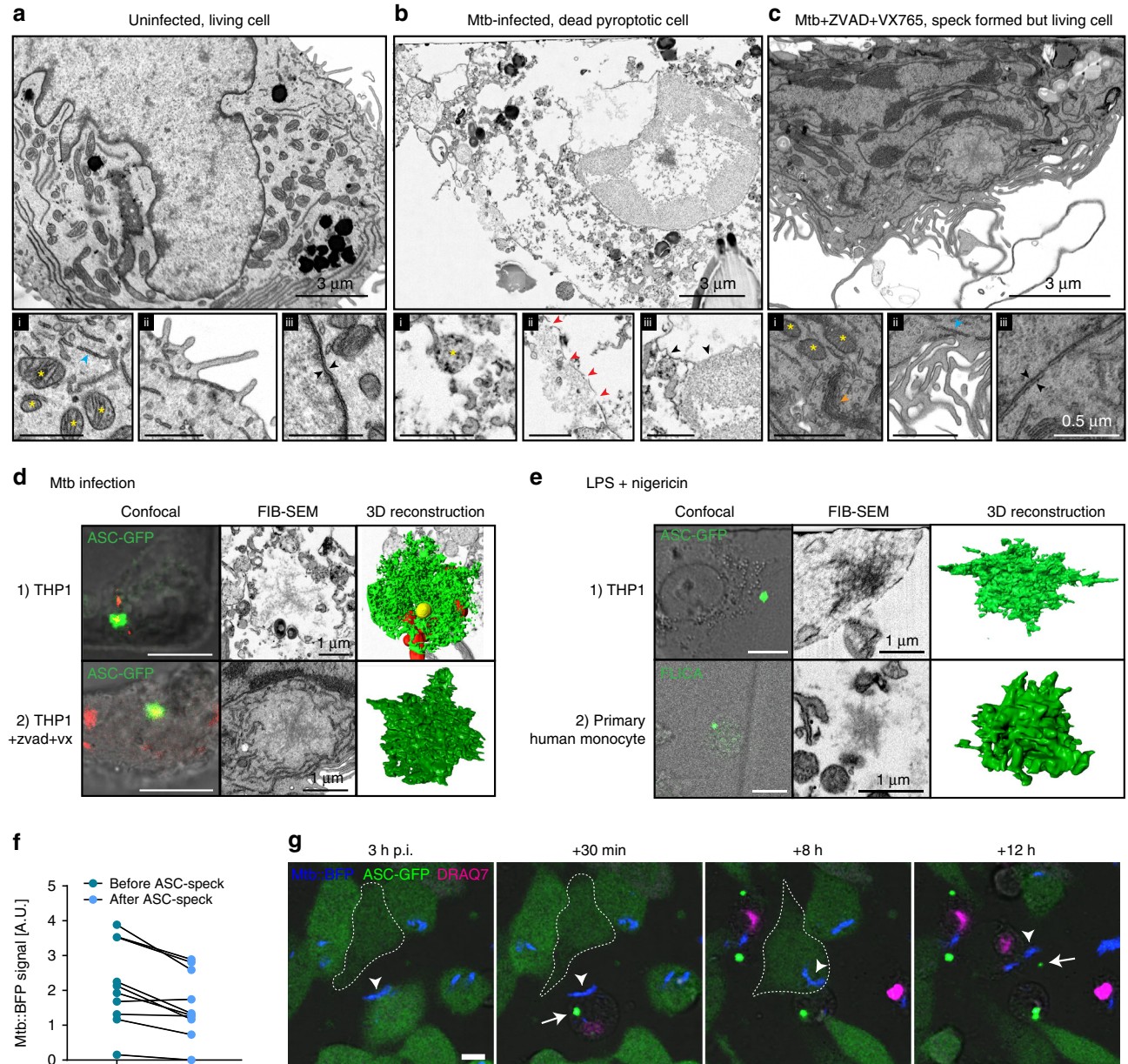

**Fig. 2 Pyroptosis causes severe cellular damage and allows for spread of Mtb.** Ultrastructural changes induced before and after inflammasome activation and cell death in cells infected by Mtb were investigated by FIB–SEM tomography. THP1-ASC-GFP cells were infected with Mtb-BFP and fixed after (**a**, **b**) 24 h or (**c**) 5 h in the absence or presence of caspase inhibitors (Z-VAD FMK and VX765), respectively. An (**a**) uninfected, (**b**) infected, pyroptotic and (**c**) infected, living cell with an activated inflammasome is displayed. The uninfected cell in (**a**) is from a 24-h infection experiment, but was found to be uninfected from confocal microscopy and FIB–SEM. Insets (i), (ii) and (iii) highlight mitochondria/ER/Golgi, plasma membrane morphology and nuclear membrane morphology, respectively. Yellow stars indicate mitochondria, blue arrowheads ER, orange arrowheads Golgi, red arrowheads plasma membrane ruptures and black arrowheads indicate nuclear membranes. Black inset scale bars represent 1 µm. Data representative of two independent experiments. **d**, **e** Structure of ASC specks induced by Mtb infection or LPS and nigericin (6,7 µM, 1–2 h) treatment in THP1-ASC-GFP cells or primary human monocytes. Monocytes were treated with FLICA caspase-1 reagent for visualisation of active caspase-1 at the inflammasome. Cells were imaged by correlative confocal microscopy and FIB–SEM, and ASC specks were reconstructed in 3D from the respective FIB–SEM image stacks. Cell numbers (1) and (2) in (**d**) are the same as in (**b**) and (**c**), respectively. Data representative of two independent experiments. **f** Quantification of intracellular Mtb-BFP fluorescence in single cells before and after ASC speck formation and pyroptosis for n = 10 representative cells. **g** THP1-ASC-GFP (green) cells in DRAQ7-containing medium (magenta) and infected with Mtb-BFP (blue) imaged by time-lapse confocal microscopy for 24 h. Arrowheads indicate Mtb that is phagocytosed twice during the time course, and arrows indicate ASC specks. Confocal scale bars: 10 µm. Source data are provided as a Source Data file.

ethyl ester (TMRE), a dye that is accumulated in the mitochondria in proportion to $\Delta\Psi_m$ (ref. [61]). We observed that $\Delta\Psi_m$ was stable in infected cells until ASC-speck formation, and that the potential quickly dropped after ASC-speck formation as measured by a drop in TMRE intensity (Fig. 3d; Supplementary

Movie 5). This immediate drop in TMRE intensity after ASC-speck formation was abolished in THP-1 GSDMD KD macrophages, without any adverse effects on the ability of the cells to form ASC specks, suggesting that GSDMD activity and possibly PM disruption during cell death are required for mitochondrial

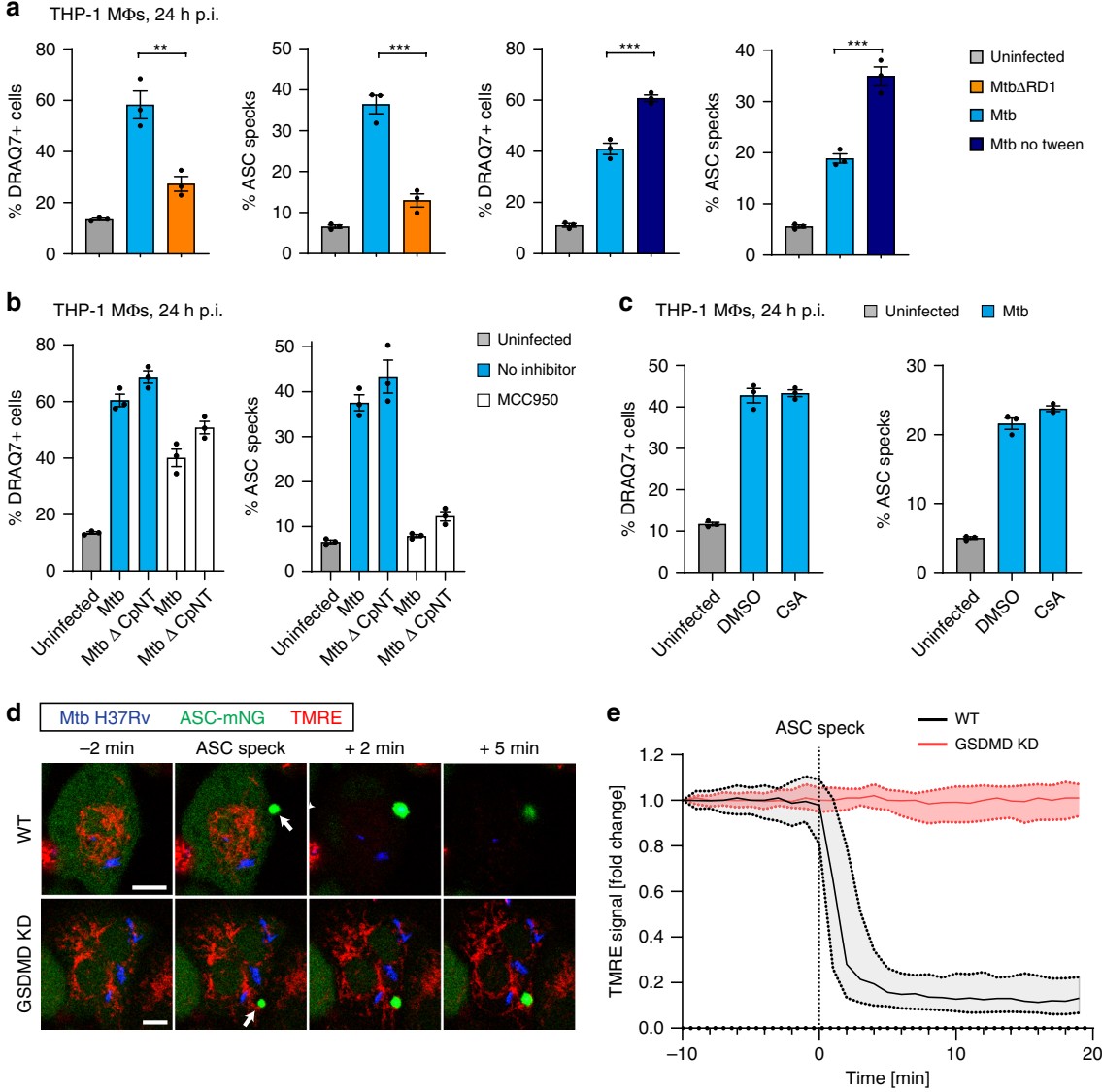

**Fig. 3 Mtb-induced inflammasome activation depends on ESX-1 but not mitochondrial damage.** In **a–c**, cells were fixed and imaged by confocal microscopy at 24 h p.i. **a** To assess the role of ESX-1 and its secreted substrates in the Mtb capsule, THP1-ASC-GFP macrophages were infected with MtbΔRD1 or Mtb cultured in the presence or absence of tween-80. **b** THP1-ASC-GFP cells were treated or not with MCC950 (as indicated), and infected with Mtb or Mtb devoid of the channel protein with necrosis-inducing toxin (MtbΔ*cpnT*). **c** THP1-ASC-GFP cells were treated with DMSO or the inhibitor of the mitochondrial permeability transition pore, cyclosporine A (CsA), and infected by Mtb. In **a–c**, ASC specks and DRAQ7+ cells were quantified for n > 2000 cells per condition, *p < 0.05, **p < 0.01, ***p < 0.001, by one-way ANOVA with two-sided Dunnett's test. Bars show mean ± s.e.m. Data representative of three independent experiments. **d** Representative time-lapse images of Mtb-BFP-infected THP1-ASC-mNeonGreen cells or THP1-ASC-mNeonGreen GSDMD KD cells, with mitochondria stained by TMRE (red). TMRE accumulates in the mitochondria in proportion to the mitochondrial membrane potential (ΔΨ$_m$), indicative of mitochondrial stability. Arrows indicate ASC specks. Scale bars 10 μm. Data representative of three independent experiments. **e** Quantification of TMRE intensity before and after ASC-speck formation in WT and GSDMD KD cells (n = 57 and n = 81 cells). Median ± IQR are shown. Data representative of two independent experiments. Source data are provided as a Source Data file.

destabilisation (Fig. 3e; Supplementary Movie 6). Similar results were obtained with LPS and nigericin treatment, where ΔΨ$_m$ dropped only after ASC-speck formation, and the drop was inhibited in GSDMD KD cells (Supplementary Fig. 3b and Supplementary Movies 7 and 8). We conclude that gross mitochondrial membrane disruption is not a prerequisite for inflammasome activation caused by Mtb or LPS and nigericin, but rather accompanies pyroptosis. These results are also consistent with the intact mitochondrial morphology observed by EM in living cells with an assembled inflammasome (Fig. 2c).

**Phagosomal damage is a prerequisite for NLRP3 activation.** One of the main effects of the ESX-1 secretion system is desta-bilisation of phagosomal membranes[35–37]. We therefore asked how phagosomal rupture is related to inflammasome activation and pyroptosis. Galectin-3 (Gal-3) binds to exposed glycosylated proteins usually confined to the inner phagosomal membrane, and has been used as a marker for ruptured phagosomes[62]. We infected THP1-Gal-3-mScarlet macrophages and observed that Gal-3 was indeed recruited to the vicinity of wild-type Mtb, but not to the vicinity of MtbΔRD1 (Fig. 4a), and Gal-3 recruitment was enhanced upon infection with Mtb grown in the absence of

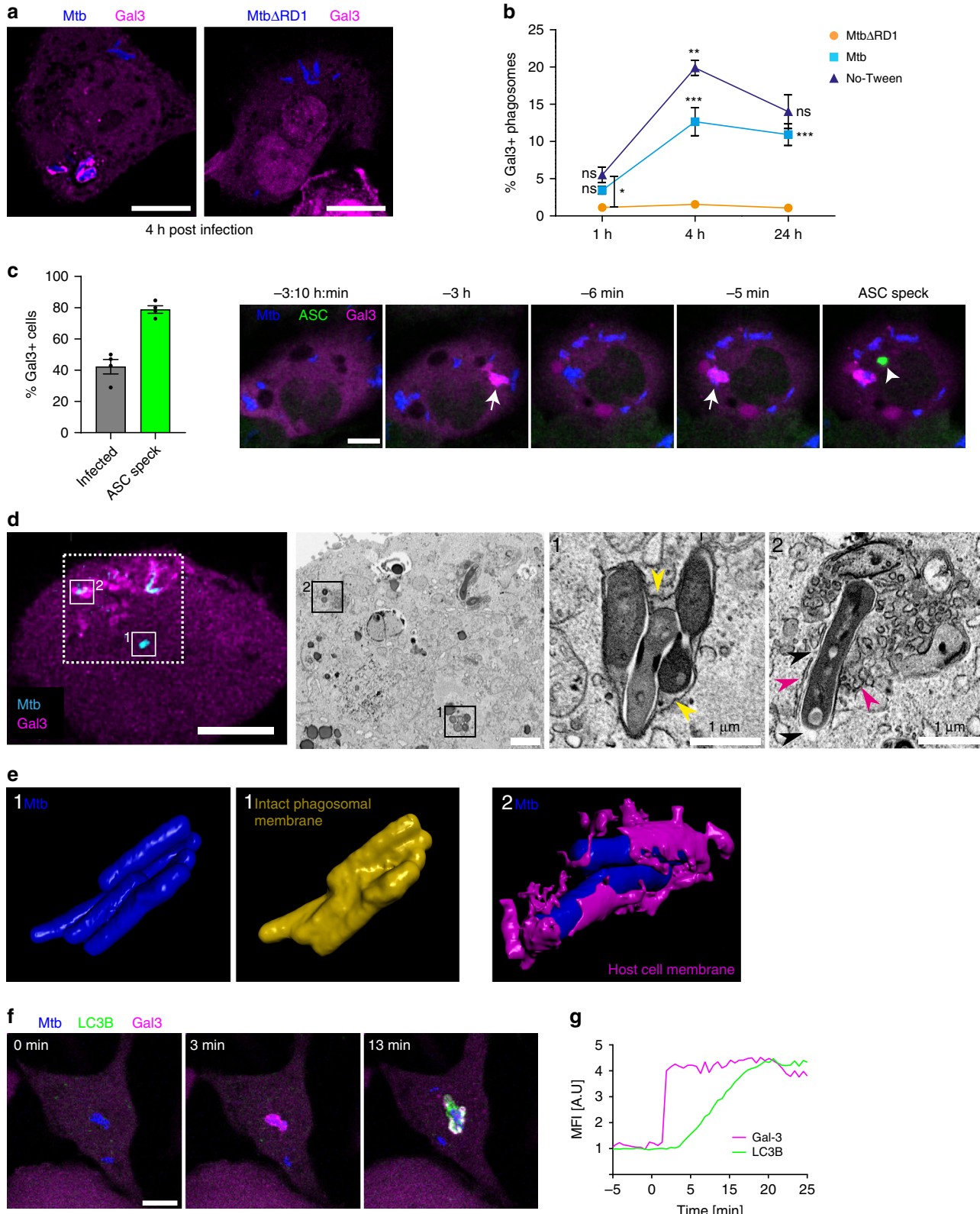

tween (Fig. 4b). Gal-3 recruitment to ruptured Mtb phagosomes occurred prior to inflammasome activation in about 80% of pyroptotic cells, and a typical sequence of events is depicted in Fig. 4c and Supplementary Movie 9.

Using CLEM, we confirmed that Gal-3 accumulation corresponds to cytosolic contact of Mtb (Fig. 4d). Mtb in phagosomes devoid of Gal-3 were surrounded by a tightly apposed and continuous phagosomal membrane, in contrast to bacteria associated with Gal-3 that had no visible phagosomal membrane, and were in direct contact with the host cell cytosol (Fig. 4d, e). Furthermore, the CLEM approach revealed in great detail the presence of clusters of vesicles and membranous structures only in regions where Gal-3 was recruited close to Mtb. The ultrastructure of Gal-3-positive compartments is similar to that

**Fig. 4 ESX-1-mediated phagosomal damage is a prerequisite for NLRP3 activation by phagosomal Mtb.** THP1-Gal3-mScarlet (magenta) cells were infected by Mtb-BFP or MtbΔRD1-BFP in the presence or absence of tween-80, fixed and imaged by confocal microscopy at the indicated time points. **a** Representative images of Gal-3+ (damaged) and Gal-3− (intact) phagosomes in cells infected with Mtb or MtbΔRD1, respectively. Data representative of three independent experiments. **b** The percentage of Gal-3+ phagosomes was quantified ($n > 4000$ phagosomes per condition, three independent experiments) by automated image analysis. Phagosomes with more than one bacterium were considered single phagosomes. $*p < 0.05$, $**p < 0.01$, $***p < 0.001$, by one-way ANOVA with two-sided Tukey's test. Statistics are calculated compared with the closest lower condition, except for the star between ΔRD1 and No-Tween at 1 h as indicated. Mean ± s.e.m. are shown. **c** THP1-ASC-GFP/Gal-3-mRuby3 cells were imaged live for 24 h, and the occurrence of Mtb-associated Gal-3 events in infected, surviving cells ($n = 221$) compared with pyroptotic cells ($n = 162$ cells) was quantified from four independent experiments. A representative time lapse showing Gal-3 (arrows) accumulation at damaged Mtb-BFP (blue) phagosomes prior to ASC-speck (green, arrowhead) formation. **d** Correlative confocal and FIB–SEM imaging of a non-pyroptotic THP1-ASC-GFP/Gal-3-mRuby3 (magenta) macrophage infected with Mtb-BFP (cyan) for 24 h. Dashed square indicates FIB–SEM area, and higher-magnification insets are numbered (1) and (2) for Gal-3− and Gal-3+ Mtb phagosomes. Yellow arrowheads indicate the intact phagosomal membrane; magenta arrowheads indicate patches of host cell membrane around a bacterium, while the absence of membrane is indicated by black arrowheads. **e** 3D reconstructions from FIB–SEM data of the bacteria from insets 1 and 2 (blue), along with the corresponding phagosomal membrane (yellow) or patches of host cell membrane (magenta). Data in (**d**), (**e**) representative of three independent experiments. **f** THP-1 cells expressing the autophagy protein LC3B-mNeonGreen (green) and Gal-3-mScarlet (magenta) were infected with Mtb-BFP (blue) and imaged by time-lapse microscopy. Data representative of two independent experiments. **g** Normalised fluorescent intensities of one representative Gal-3 and LC3B event of $n = 280$ events in two independent experiments are shown. All scale bars are 10 μm unless otherwise specified. Source data are provided as a Source Data file.

previously observed for Mtb residing in LC3+ compartments in lymphatic endothelial cells[63]. We therefore investigated the recruitment of LC3B in relation to Gal-3 by live-cell imaging of Mtb-BFP-infected THP-1 macrophages expressing mNeonGreen-LC3B and Gal-3-mScarlet. Indeed, we observed recruitment of LC3B around intracellular Mtb shortly after recruitment of Gal-3 in 98% of events (Fig. 4f, g; Supplementary Movie 10), suggesting that ruptured Mtb phagosomes are targeted by autophagy[64–66]. To investigate if autophagic targeting of Mtb leads to formation of mature, acidified autophagosomes, THP-1 macrophages expressing Gal-3-mScarlet were labelled with LysoView633 (Supplementary Fig. 4a). Only ~20% of Mtb in ruptured phagosomes was later found in acidified compartments. Together with the observation by FIB–SEM that Mtb associated with Gal-3 phagosomes retained cytosolic contact, these results suggest that autophagosomal sequestration of Mtb after phagosomal rupture is inefficient (Supplementary Fig. 4b).

**Mtb inflammasome activation is independent of pH and cathepsins.** The ability to disrupt phagosomes seems important for inflammasome activation by Mtb, and it has been reported that phagosomal acidification is a prerequisite for the membrane-damaging activity of ESX-1 (refs. [67–69]). In addition, active cathepsin release from ruptured phagolysosomes has been suggested as a trigger of NLRP3 inflammasome activation[23,40,41]. We therefore went on to investigate the possible involvement of phagosomal acidification and release of active cathepsins in inflammasome activation and pyroptosis by Mtb. Live-cell imaging of THP1-Gal-3-mScarlet or THP1-ASC-GFP cells in the presence of the pH-sensitive dye LysoView633 revealed that Mtb was equally efficient at escaping from acidified and neutral compartments (Fig. 5a–c; Supplementary Movies 11 and 12), suggesting that phagosomal acidification is not a prerequisite for Mtb escape into the cytosol. The LysoView signal in single cells was stable prior to ASC-speck formation, indicating that there is no general disruption or degradation of acidified lysosomes prior to inflammasome activation (Fig. 5d, e; Supplementary Movie 13). In addition, we did not see any effect on cell death or ASC-speck formation when cells were treated with Bafilomycin A1 (BafA1) which inhibits lysosomal acidification by V-ATPase, or the pan-cathepsin inhibitor K777, during infection (Fig. 5f), despite efficient cathepsin B inhibition (Supplementary Fig. 5). Compared with K777, cathepsin B inhibitor Ca-074-Me was less potent and had clear off-target effects on inflammasome activation during LPS and nigericin treatment (Supplementary Fig. 5).

These results demonstrate that Mtb causes inflammasome activation and pyroptosis independently of phagolysosomal acidification and release of active cathepsins.

**Mtb ESX-1 directly damages the host cell plasma membrane.** K+ efflux precedes NLRP3 activation for a range of known triggers[27]. Common for many of those triggers is that they permeabilise the PM to K+, e.g. by opening a membrane-resident pore (P2X7, pannexin-1) or by inserting and making new pores (Mixed Lineage Kinase-domain Like protein (MLKL), complement, bacterial pore-forming toxins)[70]. We and others have shown that Mtb-induced NLRP3 inflammasome activation is dependent on K+ efflux, and it has also been shown that Mtb ESX-1 can be haemolytic[71]. Moreover, PM damage has been observed during Mtb infection, and PM repair protects macrophages from necrotic cell death[72]. However, whether Mtb could directly cause PM damage or if PM damage is a consequence of Mtb-induced intrinsic cell death pathways, was not addressed. To monitor PM integrity, we made THP-1 reporter cell lines with an mNeonGreen fluorescent protein tag on the Ca$^{2+}$-binding protein Apoptosis-Linked Gene 2 (ALG-2), which is recruited to sites of PM damage by calcium influx[73–75]. We imaged Mtb-BFP-infected THP-1 macrophages by time-lapse confocal microscopy and observed ALG-2 recruitment in infected cells, almost 90% of which occurred in close vicinity to Mtb bacteria (Fig. 6a, b; Supplementary Movies 14 and 15). Mtb infection also caused ALG-2 recruitment close to Mtb in primary human macrophages (Supplementary Fig. 6a). We observed ALG-2 recruitment near Mtb both without prior Gal-3 recruitment and subsequent to Gal-3 recruitment around Mtb, with a distribution close to 50/50 between the two categories (Fig. 6b). This result suggests that Mtb damages the host cell PM either during phagocytosis or following phagocytosis and phagosome rupture.

As damage to the PM from the cytosolic side by already-internalised bacteria has not been described before, we examined whether ALG-2 recruitment next to Gal-3+ Mtb compartments indeed occurred at the PM. Total internal reflection fluorescence (TIRF) microscopy is only sensitive to the region within ~100 nm from the substrate, meaning it specifically detects events occurring at the PM[76]. We imaged infected THP-1 macrophages with wide-field and TIRF microscopy simultaneously (Fig. 6c; Supplementary Movie 16). Since the TIRF microscope is outside the BSL3 facility, we used an Mtb auxotroph strain with intact ESX-1 function (Mtb mc$^2$6206). From wide-field imaging, we observed Gal-3 recruitment to intracellular Mtb phagosomes that

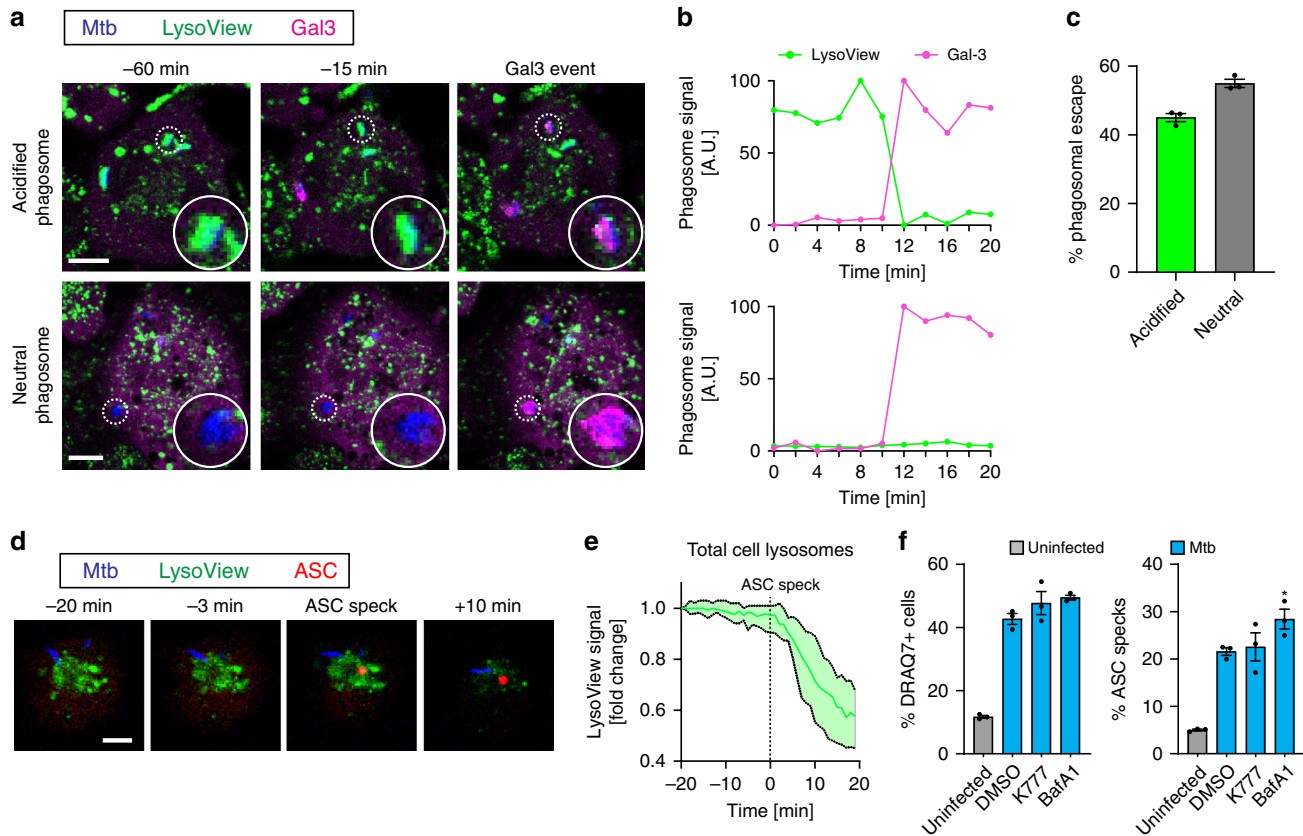

**Fig. 5 Inflammasome activation by Mtb is independent of lysosomal damage and release of active cathepsins.** THP1-Gal3-mScarlet (magenta) macrophages were labelled with a marker of acidified lysosomes (LysoView633, green), infected with Mtb-BFP (blue) and imaged by time-lapse confocal microscopy for 24 h. **a** Representative images, with circular insets showing bacteria that cause Gal-3 recruitment to their acidified or neutral compartment, respectively. **b** Traces of Gal-3 and LysoView signals for the Mtb phagosomes highlighted in (**a**). **c** Quantification of the occurrence of Gal-3 events on acidified (LysoView+) and neutral (LysoView–) Mtb phagosomes. n = 282 Mtb phagosomes analysed from 3 independent experiments. **d** Representative images of ASC-speck formation in THP1-ASC-GFP (red) cells labelled with LysoView633 (green) infected by Mtb-BFP (blue). Data representative of three independent experiments. **e** Quantification of the average LysoView signal in single cells before and after ASC-speck formation. n = 27 cells analysed, median ± IQR shown. **f** THP1-ASC-GFP macrophages were treated with DMSO, pan-cathepsin inhibitor K777 or the inhibitor of lysosomal acidification BafA1 and infected with Mtb H37Rv. After 24 h, ASC specks and DRAQ7+ cells were quantified for n > 2000 cells per condition, and the data were analysed by one-way ANOVA with two-sided Dunnett's test. Mean ± s.e.m. shown, *$p < 0.05$. Data representative of three independent experiments. All scale bars 10 μm. Source data are provided as a Source Data file.

were out of range for TIRF, while ALG-2 recruitment was observed in TIRF and wide-field modes when Mtb came into the range of the TIRF excitation, demonstrating that ALG-2 recruitment following Gal-3 also occurs at the PM.

To better understand the extent of PM damage marked by ALG-2 events, we imaged cytosolic calcium levels using the fluorescent indicator Calbryte-590 during time-lapse microscopy of Mtb-infected THP-1 cells. We observed a consistent influx of calcium when ALG-2 was recruited (Fig. 6d, e; Supplementary Movie 17), which is expected considering that ALG-2 is known to be recruited by calcium influx[73–75]. Calcium influx was not observed when Gal-3 was recruited to Mtb phagosomes (Fig. 6e; Supplementary Fig. 6b and Supplementary Movie 18). We also imaged infected THP-1 macrophages with the membrane-impermeable RNA/DNA dye propidium iodide (PI) present in the medium. We noted a consistent local influx of PI in regions of Mtb-localised ALG-2 recruitment events, in contrast to control cells with no apparent ALG-2 events (Fig. 6f, g; Supplementary Movie 19). Taken together, these data demonstrate that Mtb damages the macrophage PM when the bacteria are in close proximity. The damage caused by

Mtb makes the PM permeable to both ions ($Ca^{2+}$ and $K^+$) and larger molecules such as PI.

To investigate the ultrastructure of Mtb-associated PM damage, we performed live-cell CLEM. After live-cell imaging and rapid fixation, recent ALG-2 recruitment events were imaged by confocal or Airyscan microscopy, and further by FIB–SEM tomography (Fig. 6h; Supplementary Fig. 6c). At the location of fluorescent ALG-2 signal and directly adjacent to Mtb, multiple 50–100-nm-sized vesicles were visible, indicative of ESCRT-mediated PM repair[74,77].

**Plasma membrane damage activates the NLRP3 inflammasome.** Based on the dependency of NLRP3 activation on $K^+$ efflux as shown by us and others[27], we hypothesised that the ability of Mtb to cause PM damage and its subsequent permeabilisation to ions would be linked to inflammasome activation. We therefore imaged THP-1-ALG-2-mNeonGreen/Gal-3-mScarlet/ASC-mIRFP670 macrophages by time-lapse microscopy during infection with Mtb-BFP. An example time course is shown in Fig. 7a and Supplementary Movie 20. We recorded the time points of all Gal-3 (phagosomal rupture) and ALG-2 (PM damage) recruitment events in cells later forming ASC specks,

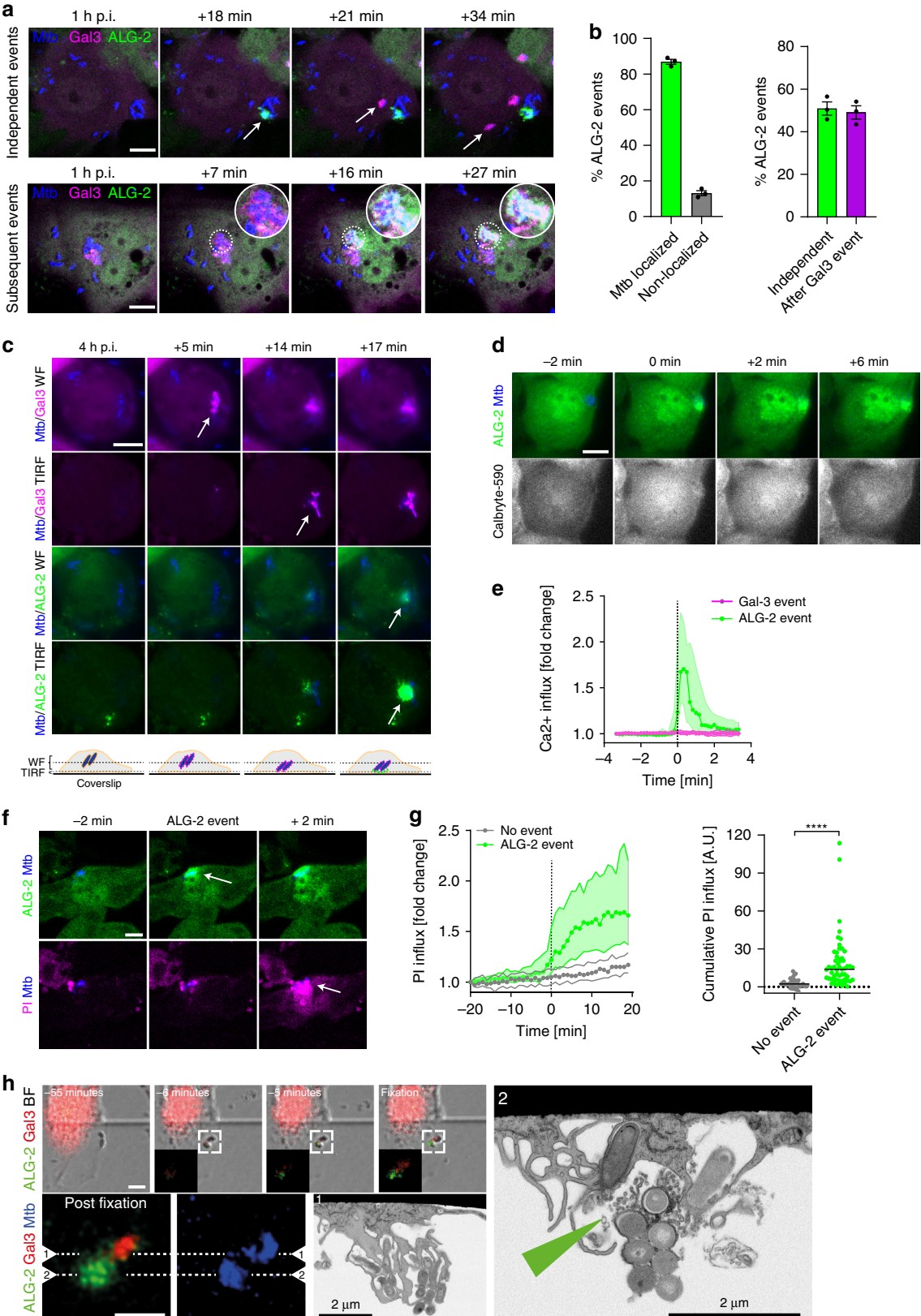

along with the time point of ASC-speck formation. PM damage preceded ASC-speck formation in 87% of cells infected with Mtb and occurred closer in time to ASC-speck formation than Gal-3 recruitment to Mtb phagosomes (Fig. 7b). While the median time from phagosomal rupture to ASC speck formation was 2 h, the median time from PM damage to ASC-speck formation was 1 h.

Moreover, within 20 min before ASC-speck formation, 48% of cells had ALG-2 events and 23% of cells had Gal-3 events, compared with 1% and 4%, respectively, in infected control cells over an average 20-min period (Fig. 7c). Thus, especially PM damage events occur much more frequently before ASC specks form, connecting these events at the single-cell level. When cells

**Fig. 6 Mtb-carrying ESX-1 can directly damage the host cell plasma membrane.** THP-1 macrophages expressing Gal-3 (magenta) and/or ALG-2 (green) were infected with Mtb-BFP (blue) and imaged by time-lapse microscopy and FIB–SEM. **a** Time-lapse images showing ALG-2 recruitment to PM-Mtb contact points either independent of or after Gal-3 recruitment. Arrows point to recent Gal-3 or ALG-2 events, and circular insets highlight a bacterium that first ruptures its phagosome (Gal-3 recruitment) and then damages the PM (ALG-2 recruitment). Data representative of three independent experiments. **b** Quantification of localisation of ALG-2 events with respect to Mtb, and of ALG-2 events occurring independent of or after Gal-3 recruitment to ruptured Mtb phagosomes. $n = 213$ and $n = 321$ events from three independent experiments. Mean ± s.e.m. shown. **c** Simultaneous wide-field (WF, cell interior) and TIRF (plasma membrane) time-lapse imaging of ALG-2 and Gal-3 during Mtb infection. Representative of $n > 10$ experiments. To verify that ALG-2 recruitment is indicative of PM damage, $Ca^{2+}$ and propidium iodide (PI) fluxes were monitored during Mtb infection. **d** Time lapse of $Ca^{2+}$ influx (Calbryte-590, grey) upon ALG-2 recruitment close to Mtb. Data representative of five independent experiments. **e** Calbryte-590 signal over time during ALG-2 or Gal-3 recruitment ($n = 10$ events per condition). Median ± IQR shown. **f** Time lapse of PI influx during Mtb-localised ALG-2 recruitment. Arrows point to ALG-2 recruitment and PI influx events occurring subsequently in the same region of the cell. Data representative of two independent experiments. **g** PI signal and the cumulative PI influx in single cells during ALG-2 events ($n = 55$) compared with neighbouring cells without ALG-2 events ($n = 37$). Median ± IQR and median values shown of all PI influx events from one experiment. Data representative of two independent experiments. ****$p < 0.0001$ by two-sided Mann–Whitney test. **h** Correlative imaging of a recent Mtb-localised ALG-2 event identified from time-lapse imaging (top row, insets). Following fixation, cells were re-imaged by confocal and FIB–SEM. SEM images (1, 2) are perpendicular to the locations indicated by dashed lines in the confocal images. Green arrow indicates possible ESCRT-associated vesicles. Confocal scale bars: 10 µm. Data representative of two independent experiments. Source data are provided as a Source Data file.

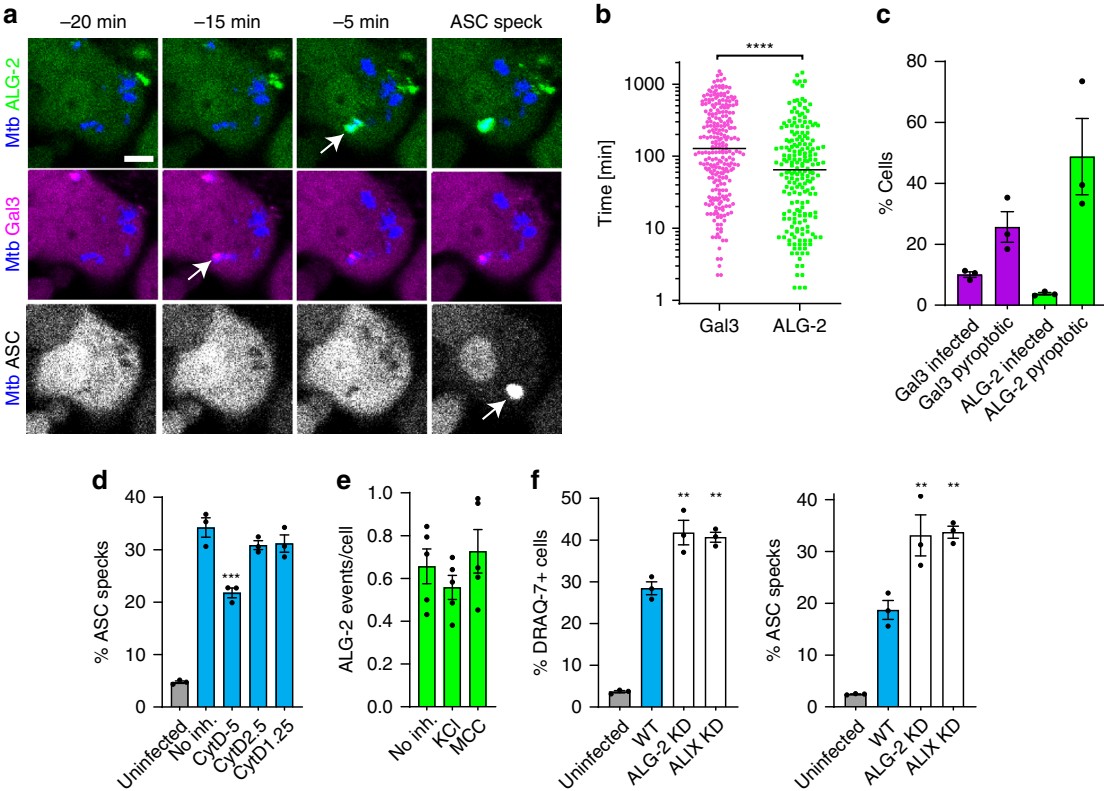

**Fig. 7 Plasma membrane damage by Mtb activates the NLRP3 inflammasome.** THP-1 cells with ALG-2 (green), Gal-3 (magenta) and ASC (grey) were infected by Mtb-BFP (blue) and imaged live for up to 24 h. **a** Time lapse of ALG-2, Gal-3 and ASC dynamics during Mtb infection, with the respective events indicated by arrows. Scale bar 10 µm. Data representative of three independent experiments. **b** Timing of Mtb-localised Gal-3 and ALG-2 events compared with ASC-speck formation. $n = 200$ ALG-2 and $n = 262$ Gal-3 events analysed from three independent experiments. Lines indicate median values. ****$p < 0.0001$ by two-sided Mann–Whitney test. **c** Percentage of pyroptotic cells with Gal-3 and ALG-2 events within 20 min before ASC-speck formation, compared with the percentage of cells in the surviving, infected population with events in an average 20-min period. $n = 104$ pyroptotic and $n = 137$ surviving cells from three independent experiments. **d** Percentage of ASC specks 24 h p.i. in cells treated with the indicated concentration (in µM) of cytochalasin D. ***$p < 0.001$ by one-way ANOVA with two-sided Dunnett's test. $n > 2000$ cells per condition in triplicate, representative of two independent experiments. Mean ± s.e.m. shown. **e** Average number of Mtb-localised ALG-2 events per cell during 24 h of infection in the absence or presence of NLRP3 inhibitors. $n = 732$ events from three independent experiments, mean ± s.e.m. of five fields of view shown. **f** Quantification of cell death and ASC specks 24 h p.i. in THP1-ASC-mNeonGreen (WT) cells and in cells depleted of the ESCRT-associated proteins ALG-2 or ALIX. Mean ± s.e.m. shown. $n > 2000$ cells per condition in triplicate, *$p < 0.05$, **$p < 0.01$ by one-way ANOVA with two-sided Dunnett's test. Data representative of three independent experiments. Source data are provided as a Source Data file.

were infected with MtbΔRD1, which does not damage host cell membranes, we did not observe any ALG-2 recruitment, and ASC speck formation was significantly reduced, as noted earlier (Fig. 3a). Treatment with the phagocytosis inhibitor cytochalasin D during infection with Mtb only partially inhibited ASC-speck formation after 24 h, despite strong inhibition of Mtb uptake, supporting the conclusion that Mtb also can activate the NLRP3 inflammasome through PM damage from outside of the cell (Fig. 7d; Supplementary Fig. 7a). Furthermore, when inflammasome activation was inhibited by MCC950 or extracellular KCl during infection, the number of ALG-2 events per cell remained constant while ASC-speck formation was inhibited, confirming that PM damage happens prior to NLRP3 inflammasome activation (and is not caused e.g. by active GSDMD) (Fig. 7e; Supplementary Fig. 7b).

ALG-2 and the ALG-2-interacting protein X (ALIX) are early components of ESCRT-mediated PM repair[73,74]. We went on to inhibit PM repair by knocking down ALG-2 or ALIX by CRISPR–Cas9 in THP-1 macrophages. Mtb caused significantly more inflammasome activation and pyroptosis both in ALG-2 and ALIX KD macrophages after 24 h of infection, compared with control cells (Fig. 7f; Supplementary Fig. 7c). Together, these results demonstrate that Mtb-induced PM damage is a trigger of NLRP3 inflammasome activation in macrophages, and that PM repair mechanisms suppress inflammasome activation and pyroptosis caused by Mtb.

Silica crystals cause silicosis, a disease with severe lung pathology, and a known risk factor for tuberculosis[23,78]. Silica activates the NLRP3 inflammasome, which has been associated with phagolysosomal damage[23]. We decided to test if PM damage is involved in triggering NLRP3 activation by silica crystals in a similar manner as with Mtb. TIRF and wide-field microscopy of THP-1 macrophages revealed Gal-3 recruitment to intracellular silica crystals and ALG-2 recruitment to sites at the PM (Fig. 8a; Supplementary Movie 21), indicating that silica crystals are also capable of disrupting phagosomal membranes and the PM. A typical time course of events culminating in ASC speck formation is shown in Fig. 8b and Supplementary Movie 22. PM damage preceded ASC-speck formation after silica stimulation in most cells (90%). Further, silica-induced ALG-2 events were closer in time to ASC specks than Gal-3 events, similar to what we observed during Mtb infection (Fig. 8c). In total, 71% of cells that assembled ASC specks had an ALG-2 recruitment event within the last 20 min before ASC speck formation, while ~25% of the cells had a Gal-3 recruitment event within the same period (Fig. 8d). Nigericin, which directly depletes the cell of $K^+$ as a $K^+$ ionophore, or imiquimod, which is an NLRP3 trigger acting independently of $K^+$ efflux[28], did not result in any visible Gal-3 or ALG-2 recruitment events prior to inflammasome activation (Supplementary Fig. 7d, e and Supplementary Movies 23 and 24).

Further, we investigated if direct damage to lysosomes without PM damage could activate NLRP3. For this purpose, we used the lysosome-targeted photosensitiser $TPCS_{2a}$[79] (Supplementary Fig. 7f, g and Supplementary Movie 25). However, despite loss of lysosomal pH and Gal-3 recruitment indicating lysosomal damage after blue-light excitation in the presence of $TPCS_{2a}$, we did not observe ALG-2 events at the PM or ASC-speck formation. These data show that PM damage (with subsequent $K^+$ efflux) is a central event upstream of NLRP3 inflammasome activation by Mtb and silica, and that lysosomal damage itself is not sufficient to activate NLRP3.

## Discussion

Both AIM2 and NLRP3 have been implicated in IL-1β release during Mtb infection of macrophages[14–16,39], while the occurrence of pyroptosis as a distinct route of cell death has been less clear[15,48,49,80]. Here we establish that ESX-1-mediated PM damage causes $K^+$ efflux, NLRP3 activation and subsequent caspase-1-mediated IL-1β release in THP-1 cells and human primary monocytes and macrophages. We also demonstrate at the single-cell level that NLRP3 activation causes rapid pyroptosis through caspase-1 and GSDMD in THP-1 cells. We emphasise that pyroptosis is an all-or-nothing response after ASC-speck formation, consistent with previous reports[81]. In primary human monocytes and macrophages, NLRP3 inhibitors reduced cell death, although the dependence on caspase-1 was less clear for macrophages. It has previously been reported that monocytes require less stimulation than macrophages to activate an NLRP3-mediated inflammatory response[82], and PMA differentiation of THP-1 cells upregulates pro-inflammatory cytokines and NLRP3 pathway components, thus potentially making them more prone to activation of this pathway[83]. In a recent report, even in response to the canonical NLRP3 trigger LPS and nigericin, a low proportion (<5%) of human macrophages formed ASC specks, ascribed to a regulatory role of alternative NLRP3 isoforms[84]. Accordingly, the poorer inhibition of pyroptosis in macrophages than in monocytes/THP-1s is likely due to a lower frequency of macrophages responding with inflammasome activation and IL-1β release[84], which are the cells prone to undergo pyroptosis.

Furthermore, caspase inhibitors have differing efficacies in preventing pyroptosis, despite potent inhibition of IL-1β release[85], and there is also crosstalk between cell death pathways and caspase redundancy during inflammasome activation[86–89]. Together with long time courses of infection, these factors could contribute to why caspase-1 inhibitors are less efficient than NLRP3 inhibition in preventing primary cell death, which is consistent with previous reports of caspase-independent cell death during Mtb infection[48,49].

We show at the single-cell level that a high bacterial burden is linked to NLRP3 activation, and other groups have also shown the correlation between bacterial burden and cell death[49,90,91]. One could hypothesise that pyroptosis would occur in vivo in tissue regions where Mtb replication results in high bacterial loads, such as in necrotising granulomas during active phases of the disease when Mtb needs to spread[91]. Further, although IL-1β is a central cytokine in a successful immune response against tuberculosis[92], a gain-of-function mutation in NLRP3 has been associated with poor clinical outcome in tuberculosis patients[93]. Thus, further studies in relevant in vivo systems will be necessary to clarify the prevalence and importance of NLRP3 activation and the pyroptotic pathway during different stages of tuberculosis disease.

Necroptosis is another necrotic cell death pathway induced by Mtb, mainly caused by the secreted $NAD^+$ glycohydrolase TNT[32,33]. In some cases, there is crosstalk between necroptosis and NLRP3 activation and pyroptosis[94,95], but our results indicate that these are distinct processes during Mtb infection. We do not see an effect of necroptosis inhibitors or TNT-deficient mutants in THP-1 cells on NLRP3 activation, while in primary human monocytes and macrophages, cell death was reduced by RIPK3 inhibition, suggesting that both pathways are indeed activated in more physiological systems. The discrepancy between our results and those of Pajuelo et al.[33] with regard to cell death by TNT and necroptosis inhibitors in THP-1 cells is likely due to the longer infection time used by Pajuelo and colleagues: our analyses are mostly performed after 24 h of infection where pyroptosis is most prominent, but we also see an increase in necrotic cell death at later time points. Mitochondrial damage and depolarisation have been linked to necrosis during Mtb infection, both necroptosis and other forms of necrosis[32,33,59]. By time-lapse imaging, we show that destabilisation of mitochondria

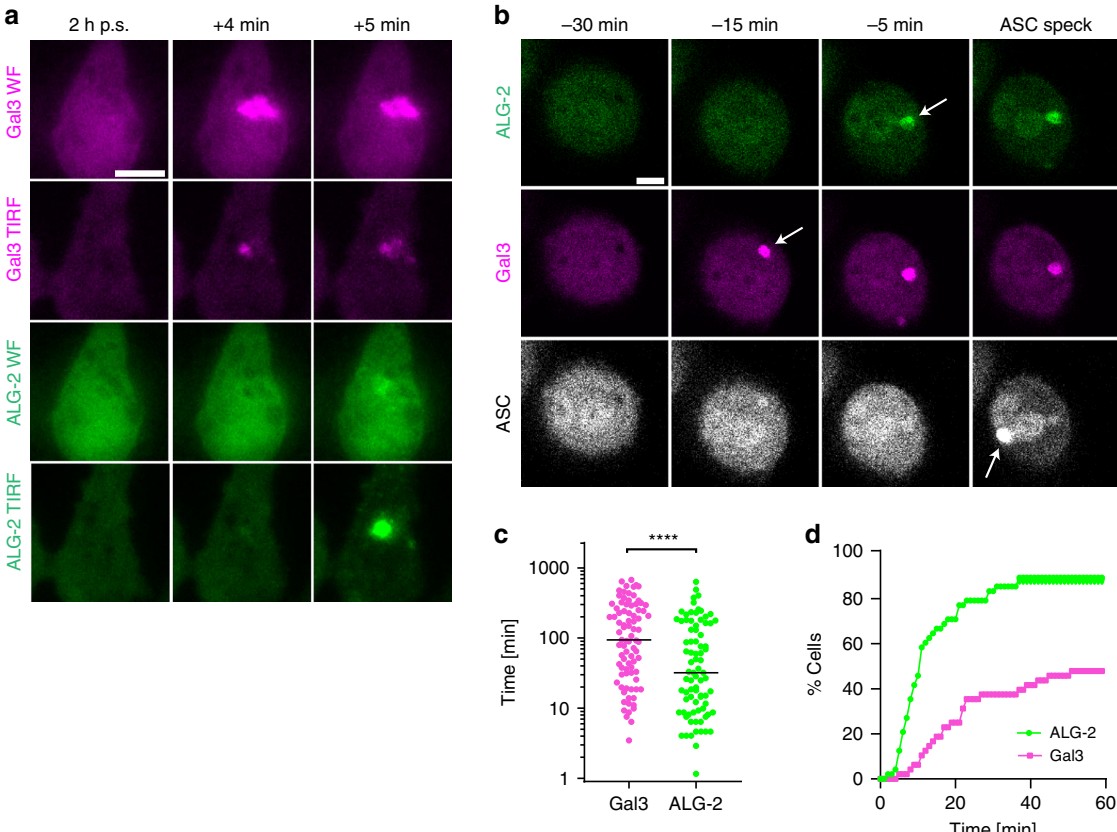

**Fig. 8 Plasma membrane damage by silica activates the NLRP3 inflammasome.** THP-1 cells with ALG-2 (green), Gal-3 (magenta) and ASC (grey) were infected by Mtb-BFP (blue) and imaged live for up to 24 h. **a** Representative time-lapse images of ALG-2 and Gal-3 from wide-field (WF) and TIRF imaging during treatment with silica (time indicated post addition of silica, p.s.). **b** Representative time-lapse images of ASC-speck formation during treatment with silica. Arrows indicate ALG-2, Gal-3 and ASC speck events. **c** Timing of Gal-3 and ALG-2 events compared with ASC-speck formation after silica treatment. $n = 82$ ALG-2 and $n = 85$ Gal-3 events analysed from three independent experiments. Lines indicate median values. ****$p < 0.0001$ by two-sided Mann–Whitney test. **d** Quantification of the percentage of cells with ALG-2 or Gal-3 events in the given time interval before ASC-speck formation. $n = 48$ cells from three independent experiments. All images are representative of three independent experiments. Scale bars 10 µm. Source data are provided as a Source Data file.

accompanies pyroptosis rather than causing inflammasome activation in the NLRP3 pathway. In summary, our results highlight that Mtb-induced pyroptosis is an independent pathway distinct from other programmed necrotic pathways such as necroptosis, although both pathways and others can and likely do occur during Mtb infection in vivo. Which pathway is triggered first is likely determined both by the state of the host cell (e.g. activation state, immune environment or previous stimulation) and the nature of the damage by Mtb, where more severe damage causes $K^+$ efflux and pyroptosis, while lesser damage could allow, e.g. TNT secretion and subsequent activation of necroptosis.

The precise mechanism of NLRP3 activation has evaded the community for over a decade. Although alternative routes were proposed, inhibition of NLRP3 activation by high concentrations of extracellular KCl positions $K^+$ efflux as the predominant upstream effector of NLRP3 activation. This is also the case for Mtb, as we and others have shown[40,96]. The similarities in ASC-speck structure and progression of pyroptosis after NLRP3 activation by Mtb infection or nigericin, which is a $K^+/H^+$ ionophore, suggest that differences are linked to the mechanism of $K^+$ efflux. Lysosomal damaging agents are one of the most clinically relevant classes of NLRP3 triggers[97]. Cathepsin release from damaged lysosomes, especially of Cathepsin B, is one commonly proposed connection, although how this further causes $K^+$ efflux has been unclear[23,40,41]. A main source of this confusion is the widespread use of the Cathepsin B inhibitor

Ca-074-Me, which appears to inhibit NLRP3 activation independent of Cathepsin B inhibition[27,42–44] (Supplementary Fig. 5). We show that although Mtb can damage acidified phagolysosomes, this is not required for NLRP3 activation, and there is no general loss of lysosomal content prior to NLRP3 inflammasome activation. Instead, PM damage appears to be the key event to trigger NLRP3 during Mtb infection. Also, in response to silica crystals, a canonical lysosome-damaging NLRP3 trigger, we observe PM damage prior to NLRP3 activation. PM permeabilisation can cause ion fluxes down their electrochemical gradients and thus activate NLRP3 by $K^+$ efflux[70].

PM damage during phagocytosis of Mtb and silica explains some of the NLRP3 activation events, but our imaging data show that PM damage can also occur from the cytosolic side after uptake, if Mtb or silica damage the phagosome and subsequently come into contact with the PM. This mechanism explains how something that primarily disrupts phagolysosomes can cause $K^+$ efflux and NLRP3 activation. Strikingly, localised and controlled lysosomal damage caused by a lysosomal photosensitiser will not subsequently damage the PM and did not cause NLRP3 activation. Since PM damage is actively repaired through many routes, including the $Ca^{2+}$-dependent ESCRT machinery[45,73–75], it is also interesting to note that this branch of ESCRT emerges as a suppressor of NLRP3 activation and cell death during Mtb infection. The ESCRT machinery was also recently shown to be involved in repair of phagosomes damaged by mycobacteria[98,99],

and to regulate pyroptosis or necroptosis by repairing GSDMD[100] or MLKL pores[101], respectively. These observations further underline the importance of the ESCRT machinery in regulating inflammation and maintaining cell viability.

How ESX-1 damages membranes is not well understood. Several papers indicate a central role for the secreted protein ESAT-6, acting through membrane lysis or membrane pore formation[34,67,69,102,103]. However, this has recently been attributed to detergent contamination, and a contact-induced form of gross membrane disruption caused by undefined ESX-1 activity has been proposed[71]. ESX-1-secreted substrates are also retained in the mycobacterial capsule, and ESAT-6 has been observed to quickly disappear upon contact between bacteria and host macrophages[54,55]. Our results also indicate a contact-induced damage, as damage events are localised to contact points between Mtb and the PM. The damage upon contact could be caused by ESX-1-dependent factors in the bacterial capsule or other factors that are diminished by the presence of the detergent tween, as Mtb cultured without tween showed increased membrane-damaging activity and higher inflammasome activation and cell death. The damage also appears to be severe, as shown by ultrastructural investigations, the simultaneous recruitment of Gal-3 and loss of lysosomal pH in phagolysosomes and the simultaneous recruitment of ALG-2 and PI and calcium influx during PM damage. Furthermore, the membrane-damaging activity of ESX-1 is unique in that it can act from both sides of the PM. Our ultrastructural investigations of sites of PM damage close to Mtb indicate that the PM repair response in the affected cell is quickly initiated, complicating a direct observation of the initial damage.

Mtb needs to escape from host macrophages to efficiently replicate and spread to new hosts. Based on our data, we propose a model where preservation of membrane integrity is key to contain Mtb infections. The ESCRT machinery repairs damages caused by Mtb to the PM and phagosomes, whereas autophagy sequesters ruptured Mtb phagosomes. Inefficient counter-measures by the host cell allow for $K^+$ efflux and NLRP3 inflammasome activation, in which case pyroptosis is inevitable and facilitates the spread of Mtb to neighbouring cells and eventually to new hosts. As shown here and elsewhere[32,33,59,104], multiple forms of programmed necrotic pathways, including pyroptosis, necroptosis and others occur during Mtb infection. Mtb likely exploits these multiple pathways simultaneously to spread within a host. Cell necrosis pathways, including pyroptosis, are thus attractive as possible targets for host-directed therapeutic strategies by limiting both bacterial spread and excessive inflammation[105].

We also show that PM damage rather than phagolysosomal damage is a shared mechanism with the crystalline NLRP3 trigger silica. Crystalline triggers cause NLRP3-driven sterile inflammatory diseases in the lungs and the cardiovascular system[97]. In particular, silicosis is a known risk factor for tuberculosis, and studies have shown both increased phagocytosis and impaired macrophage function upon co-exposure to silica and Mtb[78]. Due to the similar progression of phagosomal and PM damage during Mtb and silica exposure, we hypothesise that these processes could be exacerbated during simultaneous exposure, driving pathology. Overall, our findings point to PM damage as a shared mechanism for this group of NLRP3 activators, extending the relevance beyond tuberculosis disease.

## Methods

**Reagents**. All reagents used in this paper are listed in Supplementary Table 1.

**Experimental models**. THP-1 cells (ATCC Cat#TIB-202, ASC-GFP from Prof. Emad Alnemri) were maintained in RPMI 1640 supplemented with L-glutamine,

10 mM HEPES and 10% FCS, and passaged regularly to keep the cell density between 0.2 and $1 \times 10^6$/mL. Cell lines were routinely tested for mycoplasma. Human peripheral blood mononuclear cells (PBMCs) were isolated with Lymphoprep from buffy coats obtained from healthy volunteers (both male and female) at the blood bank of St. Olav's Hospital (Trondheim, Norway), or directly from the blood of healthy volunteers (both male and female) after informed consent. Collection of human blood was approved by the Regional Committee for Medical and Health Research Ethics in Central Norway.

Mtb strains were grown at 37 °C in Middlebrook 7H9 medium supplemented with 0.2% glycerol, 0.05% Tween-80 and oleic acid, albumin, dextrose and catalase (OADC). For Mtb mc²6206 auxotroph strain, 50 μg/mL L-leucine and 24 μg/mL D-pantothenate were added to the growth medium. Mtb bacteria were grown in the absence of Tween-80 for 24 h prior to infection in experiments where no-tween growth conditions are indicated.

Lentiviral production was performed in HEK293T cells using 3rd generation packaging system with pMDL.g/pRRE, pMD2.G and pRSV-Rev and JetPrime (PolyPlus) as transfection reagents according to the manufacturer's instructions[106]. Virus-containing supernatant was harvested 2 and 3 days post transfection, and filtered through 0.45-μm PES membrane filters. THP-1 cells were transduced in the supernatant by 90 min of spinoculation at 32 °C, 1000 g in the presence of 8 μg/mL polybrene, and selected in 1 μg/mL puromycin, 10 μg/mL blasticidine or 100 μg/mL hygromycin for 1 week. Finally, the cells were selected for a consistent moderate expression level of fluorescent constructs by 1–2 rounds of FACS (BD Aria II). CRISPR-modified THP-1 cells were used directly as a polyclonal population after puromycin selection. Mtb H37Rv, Mtb H37RvΔRD1 and Mtb mc²6206 were transformed with msp12::EBFP2 and selected on hygromycin (65 μg/mL) 7H10 plates (Difco/Becton Dickinson).

**Generation of new vectors**. Gateway cloning was used to generate lentiviral expression vectors. Human Galectin-3, human ALG-2 (THP-1 cDNA), human ASC, human LC3B (synthetic GeneArt Strings, ThermoFisher), mScarlet (synthetic), mRuby3, mNeonGreen, mIRFP670 and SNAP$_f$ tag were PCR-amplified and cloned into Gateway pEntry vectors. All pEntry constructs were verified by Sanger sequencing (GATC Biotech) before further use. PuroR in pLex307 was replaced with BlastR or HygR to generate Gateway lentiviral destination vectors with the corresponding antibiotic resistances. Two-fragment Gateway recombination with pLex307, pLex307-Blast or pLex307-Hyg was performed to generate ASC-mNeonGreen, ASC-mIRFP670, Galectin-3-mScarlet, Galectin-3-SNAP, mNeonGreen-LC3B and mNeonGreen-ALG-2 constructs. Guide RNAs targeting GSDMD, NLRP3, ALG-2 or ALIX were cloned into LentiCrispr v2 by BsmBI digestion and ligation.

**Verification of CRISPR knockdown**. Knockdown in the cell pool was verified by cleavage assay (GeneArt genomic cleavage assay, ThermoFisher), sequencing and TIDE analysis[107] and/or western blotting. Uncropped and unprocessed western blot scans are available in the Source Data File. Antibodies and dilutions for western blot were NLRP3 (1:1000), GSDMD (1:1000), ALG-2 (1:500) and ALIX (1:1000).

**Macrophage and monocyte infection and stimulation**. Before use, THP-1 cells were differentiated at a concentration of 300.000 cells/mL in medium containing 100 ng/mL phorbol 12-myristate 13-acetate (PMA) for 3 days, washed in cell medium and rested 1 day prior to experiments. Primary human monocytes were selected by plastic adherence of PBMCs for 1 h, followed by 3× washing in HBSS, and cultured in RPMI with 10% A+ serum (blood bank of St Olavs Hospital, Trondheim, Norway). For primary macrophage experiments, primary monocytes were selected by α-CD14 bead isolation (Miltenyi Biotech) and seeded out in RPMI with 5% A+ serum and 50 ng/ml rhGM-CSF (Peprotech, Gibco–ThermoFisher) for 3 days. At the third day, the media was replaced with normal cell medium with 5% A+ serum.

Mtb bacteria were grown until OD$_{600}$ was 0.4–0.55 (log phase), then pelleted at 2400 g for 10 min, resuspended in RPMI with 10% A + serum to opsonise bacteria prior to infection and sonicated 2–3 times for 5 s at 70% power (Branson Digital Sonifier, S-450D). Clumped bacteria were removed by centrifuging at 300 g for 4 min. The supernatant containing bacteria was diluted in RPMI with 10% human A+ serum to MOI 20, unless otherwise indicated, assuming 1 OD$_{600}$ = $3 \times 10^8$ bacteria/mL. For primary human macrophage experiments, the second centrifugation was done at 200 g for 1 min, and bacteria were resuspended in 5% A + serum (to reduce the background of the LDH readout). Bacteria were applied to cells for 45 min (THP-1 and monocytes) or 4 h (primary human macrophages) with or without inhibitors, followed by washing in HBSS and replacement of the media to normal cell medium with 10% or 5% A+ serum with or without inhibitors, or Leibovitz L-15 CO$_2$-independent medium with 10% A+ serum for live-cell imaging. The typical condition of MOI 20 infection gave ~50% infection rate with 1–20 bacteria per macrophage. Inhibitor concentrations were DMSO control (1:400), z-VAD-FMK (50 μM for THP-1 cells, 20 μM for primary human cells), VX765 (50 μM), MCC950 (10 μM), KCl (40 mM), Nec1s (10 μM), GSK-872 (5 μM), Cyclosporin A (5 μM), K777 (15 μM) or Bafilomycin A1 (50 nM). Supernatants were harvested for ELISA or LDH assays after 24 h, and analysed by

human IL-1β kit or LDH cytotoxicity kit according to the manufacturer's instructions.

In stimulation experiments, the following conditions were used: LPS (10 ng/ml LPS, 3 h), FSL-1 (10 ng/ml), Pam3Cys4K (10 ng/ml), nigericin (10 μM, 1 h, unless otherwise stated), imiquimod (20 μg/ml, 1 h) and silica (100–200 μg/ml). When used in combination, inhibitors were added 30 min before and were present during stimulation.

**Live-/fixed-cell imaging.** THP-1 cells were seeded in 35-mm glass-bottom dishes (Ibidi) or 96-well glass-bottom plates (Cellvis) as above. For some experiments, bacteria were added at a lower MOI and not washed to be able to follow the entire course of infection. Cells were imaged on a Leica SP8 confocal microscope with a 37 °C incubator using a 40 × 1.3 or 20 × 0.75 oil immersion objective or 10 × 0.4 air objective at the time points indicated in the paper. TIRF/wide-field imaging was done on a Zeiss Laser TIRF 3, with a 63 × 1.46 oil objective and Hamamatsu EMX2 EMCCD camera. For quantification of cell death (by DRAQ7 signal) and ASC specks at defined time points with and without inhibitors or knockdown, 16 fields of view containing $n > 2000$ cells in total were imaged per condition in triplicate. For time-lapse imaging by confocal microscopy, typically six fields of view comprising >500 cells in total were imaged at 30–45-s intervals, while TIRF/WF was done on one to four fields of view comprising 10–40 cells at 10–30-s intervals. Fields of view were defined based on plate coordinates, without bias from observation of the sample, and all cells or events within every captured field of view were included in the analysis. For some experiments, DRAQ7 (0.15 μM), TMRE (10 nM), LysoView633 (1:10 000) or Calbryte-590 (5 μM) was preincubated for 30–60 min and added to the cell medium during Mtb infection. Calbryte-590 was only preincubated and not re-added. In cells with SNAP tag, cells were labelled with 1:200 SNAP-Cell 647-SiR for 15 min, washed 3× in cell medium and rested for 30 min before use. For lysosomal damage by photosensitiser, TPCS$_{2a}$ was added to the medium (0.4 μg/mL) for 16 h, and cells were washed and incubated for 4 h without TPCS$_{2a}$. THP-1 cells were stained or not stained with Cresyl Violet (1 μM, 5 min) exposed to 405-nm light (10%, 1 s), and imaged. Galectin-3 accumulation on phagosomes during infection with different Mtb strains was determined by fixing cells in 4% PFA for 20 min at 1, 4 and 24 h post infection, and imaging 16 fields of view with 40× objective per well. Caspase-1 activity was detected in monocytes using FAM-FLICA Caspase-1 Assay kit according to the manufacturer's instructions.

**Immunofluorescence analysis of ALG-2 in primary macrophages.** Primary human macrophages were isolated, differentiated and infected with Mtb auxotroph (Mtb H37Rv mc²6206::EBFP2) as previously described. Cells were fixed on ice 2 h 15 min post infection in methanol–acetone solution (50% methanol and 50% acetone). Cells were immediately stored at −20 °C. The following day, the fixative was decanted, and cold PBS was added to the cells for 1 h at room temperature. Blocking was done in PBS with 20% human serum for 20 min. Cells were incubated with ALG-2 antibody (1:50 in PBS with 20% human serum) overnight at 4 °C, washed with PBS × 3, stained with secondary antibody Alexa Fluor 647 (1:500 in PBS with 20% human serum) for 1 h and finally washed in PBS × 3. Images were acquired with Zeiss LSM880 microscope, and fluorescent images were deconvolved using Huygens Professional v18.10 (Scientific Volume Imaging, The Netherlands, http://svi.nl).

**FIB–SEM sample preparation and correlative imaging.** For correlative imaging experiments, two growth substrates were used, with different optical microscopy and fixation approaches. In the first approach, THP1-ASC-GFP Gal-3-RFP cells were seeded and infected as above on aclar slides micropatterned by thermo-moulding, using a silicon stamp produced by a photolithography workflow. Aclar substrates were placed in 24-well plates and imaged live using 10× objective as above. After 24 h, cells were fixed in 2.5% glutaraldehyde in 100 mM PIPES for 1 h, placed upside-down in 35-mm glass-bottom dishes and cells of interest were re-imaged using 63 × 1.2 W objective on a Leica SP8 confocal microscope. In the second approach, THP-1 cells were seeded and infected as above on micropatterned polymer coverslips (ibidi) mounted in 8-well sticky-Slides (ibidi). Samples were imaged using 40 × 1.3 oil objective on a Leica SP8 confocal microscope and fixed by adding 8% paraformaldehyde (PFA) and 0.2% GA in 200 mM PIPES directly into imaging media at a 1:1 ratio. After 20 min, fixation media was changed to fresh 4% PFA and 0.1% GA in 100 mM PIPES. Cells with interesting events during live microscopy were then re-imaged with a 40 × 1.2 Imm AutoCorr objective on a Zeiss LSM880 microscope using the Airyscan mode. After re-imaging, cells were placed in fresh fixative (2% PFA and 2.5% GA in 100 mM PIPES) and incubated at 4 °C overnight. After fixation, cells were postfixed and contrasted in 1.5% potassium ferrocyanide and 2% osmium tetroxide in 0.1 M cacodylate buffer for 4 h on ice, on a shaking platform. Cells were then dehydrated in 50% and 70% ethanol for 10 min per step, on a shaker. Further, cells were en bloc stained in 2% uranyl acetate in 75% ethanol for 1 h and 20 min at room temperature. Cells were further dehydrated in 90% and 100% ethanol (four steps) and embedded in Durcupan (Sigma-Aldrich) through three dilutions of ethanol and Durcupan (2:1, 1:1 and 1:2) for 1.5 h per step on a shaking platform. Samples were incubated on a rotator in 100% Durcupan

overnight, changed to fresh Durcupan the next day and cured at 60 °C for a total of 3 days. Polymer growth substrates were removed, and samples were attached to SEM sample stubs with a drop of Durcupan before the last day of curing. Before FIB–SEM imaging, a layer of 40 nm platinum/palladium was deposited on the samples using a Cressington sputter coater model 208 HR. FIB–SEM imaging and tomography experiments were performed on a Helios G4 dual-beam FIB–SEM from ThermoFisher, with the software Auto Slice and View 4. Acceleration voltage was 2 kV or 3 kV for the electron beam, and immersion mode and the in-column mirror detector were used for image acquisition. Slice thickness was typically set to 20 nm or 25 nm. Initial alignments, scaling and simple adjustments (brightness/contrast) were done in Fiji, while 3D reconstructions were performed semi-manually in Avizo or Dragonfly.

**Cathepsin B activity assay.** THP-1 cells were differentiated in 24-well plates as above. One hour prior to lysis, inhibitors (K777 and Ca-074-Me) were added at concentrations indicated in the main text. Cells were lysed in 200 μL of lysis buffer (0.1% Triton X-100, 250 mM sucrose, 20 mM HEPES, 10 mM KCl, 1.5 mM MgCl$_2$, 1 mM EDTA, 1 mM EGTA and 0.5 mM Pefabloc SC, pH=7.5) on ice for 20 min[108]. In total, 50 μL of lysate was mixed with 50 μL of Cathepsin B reaction buffer (50 mM sodium acetate, 4 mM EDTA, 0.5 mM Pefabloc SC, 8 mM dithio-threitol (DTT) and 50 μM Z-RR-AMC, pH = 6.0), incubated at 30 °C for 5 min and fluorescence intensity was monitored every 30 s for 20 min (ex 355/em 460 nm, POLARstar Omega microplate reader).

**Image analysis.** Cell death pathways (Fig. 1f) were identified according to the following guidelines: pyroptotic cells were characterised as those forming a visible ASC speck followed by rapid influx of DRAQ7, while necrotic cells did not form ASC specks and generally followed a slower progression with, e.g., cessation of cell migration prior to a slower influx of DRAQ7. Apoptotic cells were identified by a characteristic membrane blebbing and no DRAQ7 influx.

CellProfiler was used to automatically segment and count live cells and ASC specks based on the ASC-GFP signal, dead cells based on the DRAQ7 signal and Mtb uptake based on Mtb-BFP signal overlapping with ASC-GFP signal in live cells. Bacterial burden per cell was also measured from live-cell time-lapse movies by mean Mtb-BFP intensity in infected cells immediately prior to ASC speck formation, and compared with cells surviving the duration of the experiment. The time point for the control cells was chosen to be the average time point for ASC-speck induction. Traces of DRAQ7, TMRE, Calbryte-590, PI or LysoView signal were generated by manually selecting ROIs around cells with or without ASC specks forming and measuring the mean intensity in the ROI over time. The data were aligned according to the time point of the first visible sign of the event under consideration and plotted as mean ± IQR.

In fixed cells images of Galectin-3 colocalisation with Mtb, the colocalisation was scored by a trained model in Ilastik and further analysed by CellProfiler. For analysis of ALG-2 recruitment events compared with Gal-3 recruitment events, only events where a complete overlap between increased Gal-3 signal and Mtb was followed by an overlap with an increased ALG-2 signal were scored as "ALG-2 after Gal-3" events. Data were plotted and statistically analysed using Python or GraphPad Prism, while images and figures were prepared using FIJI and Adobe Illustrator. TIRF/WF images of mNeonGreen-ALG-2 and Galectin-3 mScarlet were unmixed by linear unmixing due to substantial cross-excitation of mScarlet with 488-nm laser.

**Statistical analysis.** Detailed statistical analyses for individual experiments are listed in each figure legend. This includes the statistical test performed, the parameters shown and number of cells, replicates and independent experiments as appropriate. Data comparing the mean values of technical replicates of representative or independent experiments were analysed by two-sided unpaired Student's $t$ test (two groups) or one-way ANOVA (more groups). For one-way ANOVA, corrections for multiple comparisons were applied for comparisons with one control group (Dunnett's) or all groups (Tukey's). For comparisons of non-parametric data (fluorescent intensity or event timing), Mann–Whitney $U$ test was used instead. GraphPad Prism 8.0 was used to perform all statistical analyses and determine $p$ values, with $p$ value <0.05 considered significant.

**Reporting summary.** Further information on research design is available in the Nature Research Reporting Summary linked to this article.

## Data availability
All data are available from the corresponding author upon reasonable request. The source data underlying Figs. 1, 2f, 3a–c, e, 4b, c, 5c, e, f, 6b, e, g, 7b–f, 8c–d, Supplementary Fig. 1b, c, e–i, 3a, c, 5a, b and 7a, b are provided as a Source Data file.

## Code availability
The Python scripts for aligning intensity time-trace data and the CellProfiler and Ilastik pipelines used in this study are available from the corresponding author upon reasonable request.

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

## Acknowledgements
We thank Liv Ryan and Unni Nonstad for FACS of THP-1 cells, Katherine Fitzgerald and Egil Lien at UMass for reading and commenting on the paper and members of the mycobacteria-HIV research group at CEMIR for valuable discussions. The fluorescence microscopy work was performed at the Cellular and Molecular Imaging Core Facility (CMIC), NTNU. CMIC is funded by the Faculty of Medicine at NTNU and Central Norway Regional Health Authority. Funding sources: The Research Council of Norway for support to the Norwegian Micro- and Nano-Fabrication Facility (NorFab) project number 245963/F5 (M.S.B. and S.U.); the Centre of Excellence grant to CEMIR project number 223255 and FRIPRO project number 287696 (to T.H.F.). NTNU Enabling Technologies for PhD funding project number 81771297 (to T.H.F.). The National Institutes of Health (NIH) by the grant R01 AI121354 (to M.N.).

## Author contributions

Conceptualisation: T.H.F., K.S.B. and M.S.B.; investigation: K.S.B., M.S.B., S.U., R.S., H.K., A.M., S.E.Å., M.H. and T.Å.S.; writing—initial draft: T.H.F., K.S.B., H.A.S. and M.S.B.; writing—review and editing: M.S.B., K.S.B., S.U., R.S., M.N. and T.H.F.; supervision and funding acquisition: T.H.F. All authors discussed the results and commented on the paper.

## Competing interests
The authors declare no competing interests.
