## [Peer Review File · Nature Communications]

Reviewers' comments:

Reviewer #1 (Remarks to the Author):

An interesting and extensive set of experiments on the interaction between THP-1 cells, human monocytes and *M. tuberculosis* is presented. An extensive series of control experiments are also included. The findings are very heavily dependent on a wide variety of microscopic techniques. The major novelty demonstrated is ESX-1 mediated, contact-induced plasma membrane damage response that occurs during phagocytosis or from the cytosolic side of the membrane after phagosomal rupture in tuberculosis. Plasma membrane injury causes K⁺ efflux and activation of NLRP3 dependent IL-1 β release and pyroptosis, facilitating the spread of Mtb to neighbouring cells. There is also evidence of interplay of pyroptosis with ESCRT-mediated membrane repair.

The reviewer is not skilled in the interpretation of microscopic images, particularly EM. The work appears to have been done meticulously but the reviewer is also aware that interpretation of such images in relation to tuberculosis infection has provoked controversy in the past.

Secondly from a holistic point of view the experimental system is restricted to highly-differentiated transcriptionally active THP-1 cells with confirmatory experiments in human monocytes. Monocytes are not the major target of intracellular tuberculosis, neutrophils and tissue macrophages are. No *in vivo* (either experimental or from infected humans) data to corroborate findings is presented. Indeed, it might seem unreasonable to suggest this as so much is presented already, but the limitation exists.

The second is that this is yet another instance that demonstrates an ostensibly important influence of the RD1 encoded genes on the pathogenesis of tuberculosis yet we are no further towards a mechanism such that this might be inhibited directly, rather than the downstream consequences. Where does the translational field go with this? Selective potassium channel blockade?

Thirdly it is also clear that the determinants of apoptotic, necrotic and pyroptotic cell death modes are indeed quite liable to influence progression or resolution of an infection. All three modes are demonstrated indeed in these experiments. Why do the authors think this happens? What are the critical early cellular determinants?

A few specific points

1. Although the figure legends are already quite long, I found it difficult to view them in isolation to work out what had been done and why. The emphasis in the legends is very technical mentioning MOI and drug doses rather than giving any hint what the bigger question under examination is.
2. Likewise many abbreviations are introduced from the abstract onwards without being spelt out. This again was not an aide to the non-specialist.
3. The experiments using silica were interesting in the context of a tuberculosis study. Silicosis is a major risk factor for tuberculosis in mining communities and yet no mention of this is made by the authors in their discussion. I though more could potentially have been made of this.

Reviewer #2 (Remarks to the Author):

In this manuscript by Beckwith et al, the authors demonstrate that Mtb-induced damage to the host cell plasma membrane activates the inflammasome and pyroptosis, facilitating the spread of Mtb to neighboring cells. Overall, using live cell imaging and CLEM, the author's carefully study the

sequence of events in which ESX-1 mediated contact induces plasma membrane damage from either extracellular or cytosolic bacilli; this membrane injury allows K⁺ efflux, which triggers inflammasome activation, IL-1 β release, and pyroptosis unless the membrane damage is balanced by repair mediated by the endosomal sorting complexes required for transport (ESCRT) machinery. This is a well-written, interesting study with well-controlled experiments, and the findings help clarify an important aspect of Mtb biology, as well as suggesting that plasma membrane damage is a common mechanism utilized by other NLRP3 activators that have previously been shown to act through lysosomal damage. This is an interesting and important paper, and I have only minor concerns/comments.

1. In the discussion about Gal3 recruitment to bacteria (lines 214-229), the authors suggest that Gal3 is recruited to bacteria with no visible phagosomal membrane and that the membrane that is visualized in Figure 4d and 4e (region 2) is an autophagosome. Does this mean the authors do not think Gal3 is binding to host glycans on the luminal surface of the phagosome? Is it directly binding Mtb?

2. Of the Mtb that become Gal3+, the author's indicate that only 20% are found in an acidified compartment, concluding that autophagy is inefficient. What percent become LC3+? Is it sequestration into an autophagosome that is inefficient or maturation to an autolysosome?

3. Lines 395-399: The author's might comment/acknowledge that although IL-1 β is central to a successful immune response in vivo, it is not NLRP3 dependent (in mice).

4. The authors claim that plasma membrane damage, occurring during phagocytosis or from the cytosol after phagosomal damage, is the cause of K⁺ efflux and subsequent NLRP3 activation by Mtb. In support of this, they observed ALG-2 recruitment to Mtb in association with Gal3 recruitment and independent of Gal3 recruitment in roughly equal proportions (Figure 6a-b). In Figure 6h, is the Gal3+ ALG-2-associated Mtb actually intracellular in the FIB-SEM? The image is difficult to interpret. If Gal3 can directly bind Mtb, it could bind extracellular Mtb, which might confound some of these studies.

5. There is no call out to Figure 4e in the text.

6. There are no error bars in Figure S1a.

7. The figure 6 legend does not explain the what is surrounded by dashed circles, intact circles, or indicated by arrows.

Reviewer #3 (Remarks to the Author):

In this manuscript "Plasma membrane damage causes NLRP3 activation and pyroptosis during Mycobacterium tuberculosis infection" Beckwith and colleagues investigated the activation of the NLRP3 inflammasome and role of pyroptosis in the death of human macrophages following in vitro infection with Mtb. They showed that cells containing ASC speck formation as a result of NLRP3 inflammasome activation, undergo pyroptosis since cell death was diminished when cells were deficient in NLRP3 or gasdermin D or treated with caspase inhibitors. They also describe experiments showing that Mtb induces plasma membrane damage leading to increased calcium influx and consequently ASC speck formation. The authors conclude that Mtb triggers pyroptosis through a mechanism associated with plasma membrane perturbation occurring prior to and/or after phagocytosis and that this is the major mechanism of cell death in Mtb infected human macrophages.

Overall, the manuscript is clearly written and the authors have effectively and elegantly employed

different strategies of single cell imaging analysis to support their claims. A major concern with this manuscript is that the authors performed most experiments using PMA treated THP-1 cells which are commonly used as a model of human macrophages for the study of inflammasome activation because they express high levels of NLRP3, ASC and pro-caspase-1. It is well known that PMA stimulates a time- and concentration-dependent rise in cytosolic calcium levels. In addition, PMA has been reported to induce pro-IL1b, pro-caspase-1 and NLRP3 expression and, under some conditions triggers NLRP3 inflammasome activation. Also, PMA has been shown to increase the expression of P2X7R which is important for both NLRP3 inflammasome activation as well as necrotic cell death induction by facilitating calcium influx. Although these responses will not trigger pyroptosis by themselves, they may bias the induction of a pyroptotic pathway of cellular necrosis following Mtb infection. In the absence of more extensive experiments with non-PMA stimulated primary macrophages, the authors need to seriously address these concerns/provisos in the Discussion of the manuscript

In one experiment (Figure 1D) the authors did employ human monocyte derived macrophages to support their claims of MTb induced pyroptosis induction based on their studies with PMA stimulated THP-1 cells. They observed a reduction in LDH release under conditions of caspase1 and NLRP3 inhibition. Nevertheless, this decrease in cell death was minor (roughly 10-40% of the normalized value of LDH release) leaving unaddressed the mechanism(s) responsible for the remainder of the death observed. Moreover, in contrast to the experiments involving THP-1 cells, the cell death of these human monocyte derived macrophages measured by LDH release was not confirmed by DRAQ7 dye staining.

Another confusing finding concerns the role of Gasdermin D which in contrast to caspase 1 or pan caspase or NLRP3 inhibition appears to account for all of the cell death observed in ThP1 cells. The authors fail to comment on this discrepancy and whether or not it may suggest the involvement of other death pathways. While the authors do acknowledge the possible participation of other death mechanism in their cultures (e.g. figure 1F) these are not adequately investigated considering their major role based on the relatively minor effects of the inflammasome inhibitors studied.

The authors argue that plasma membrane damage is induced both as a consequence of contact with extracellular bacteria and as result of Mtb phagocytosis. This conclusion is based on cell treatment with the phagocytosis inhibitor (cytD). Indeed, ASC speck formation was partially reduced (by 10% it appears) when cells were treated with cytD at 5uM. However, they did not independently measure the effect of the drug on Mtb uptake or test higher concentrations of CytD that have been documented previously to be more effective in preventing Mtb phagocytosis. For this reason it is difficult to assign a role for extracellular Mtb in triggering membrane damage and no further experiments were presented to examine such a mechanism/pathway.

Minor point:

In most assays performed the number of cells analyzed (n=27-82, depending on the experiment) is unusually low (less than 0.1% of the total number of cells in the culture) and may not offer a fair representation. This to me is an inherent weakness in the authors' data.

Although the manuscript presents interesting findings and the focus on human macrophages is laudable, the authors evidence for a role for pyroptosis in Mtb cell death is in disagreement with numerous studies employing murine macrophages, the papers on human monocyte derived macrophages by Welin et al. and Lee et al. 2011 they reference, as well a recently published article (Pajuelo D et al. Cell Microbiol. 2019) also employing human ThP1 cells that argues for a necroptotic mechanism of cell death. For this reason it is important that the authors experimentally strengthen their claim, more carefully rule out the involvement of additional death pathways in their in vitro system and address the controversy their findings raise.

Point-by-point response to the referees' comments

We would like to thank the reviewers for carefully reading and commenting on our manuscript, and also for constructive critics and valid questions regarding our results and conclusions. We have addressed the criticism by providing new data to substantiate our claims, clarified issues raised by the reviewers and rewritten parts of the manuscript accordingly. We have addressed the critics as follows; reviewers' comments followed by our response in blue:

Reviewer #1 (Remarks to the Author):

An interesting and extensive set of experiments on the interaction between THP-1 cells, human monocytes and *M. tuberculosis* is presented. An extensive series of control experiments are also included. The findings are very heavily dependent on a wide variety of microscopic techniques. The major novelty demonstrated is ESX-1 mediated, contact-induced plasma membrane damage response that occurs during phagocytosis or from the cytosolic side of the membrane after phagosomal rupture in tuberculosis. Plasma membrane injury causes K⁺ efflux and activation of NLRP3 dependent IL-1 β release and pyroptosis, facilitating the spread of Mtb to neighboring cells. There is also evidence of interplay of pyroptosis with ESCRT-mediated membrane repair.

The reviewer is not skilled in the interpretation of microscopic images, particularly EM. The work appears to have been done meticulously but the reviewer is also aware that interpretation of such images in relation to tuberculosis infection has provoked controversy in the past.

We appreciate the reviewers understanding of our work. We agree that interpretation of EM data regarding *M. tuberculosis* cytosolic contact has been controversial in the past. However, due to continuous improvement in EM technology, together with the identification of alternative markers for cytosolic contact and phagosomal damage, the field is now in agreement that a certain extent of Mtb to cytosolic contact occurs during infection in macrophages. In addition to the most important original references, we have also added a review (Simeone et al., PMID: 27247079) discussing this topic to the relevant section of the introduction (page 4, new reference #38).

Secondly from a holistic point of view the experimental system is restricted to highly-differentiated transcriptionally active THP-1 cells with confirmatory experiments in human monocytes. Monocytes are not the major target of intracellular tuberculosis, neutrophils and tissue macrophages are. No in vivo (either experimental or from infected humans) data to corroborate findings is presented. Indeed, it might seem unreasonable to suggest this as so much is presented already, but the limitation exists.

We agree that macrophages are the major target of Mtb in vivo. However, we would add that newly recruited monocyte-derived macrophages (RMs) are indeed an important host for Mtb during in vivo infection (Norris and Ernst, PMID: 30365557), which would more resemble THP1 cells and the primary monocyte/macrophages on which we have performed experiments. We would also like to point out that we did not use circulating monocytes in our experiments, but monocytes adhered overnight which can be considered young macrophages. To further ensure the relevance of our findings in primary cells we have compared several different protocols to differentiate macrophages and infected them with Mtb. Several durations (3-4 days and 7 days), differentiation stimuli (30% human serum, 10 ng/ml M-CSF, 10 ng/ml GM-CSF, 50 ng/ml GM-CSF), and isolation methods (adherence, CD14 beads isolation) in combination were tried. In all of the cases, we tested the effects of both 4 hours and overnight infection with the auxotrophic Mtb strain mc²6206 in the presence of 5% human serum. The % LDH release from cells from two or three donors per condition after overnight auxotroph Mtb infection are presented below:

The cells died at comparable levels after 7 days of differentiation vs. 3 days of differentiation. There was no clear difference between M-CSF and GM-CSF differentiation in terms of Mtb-induced toxicity (% LDH release).

Based on this differentiation pilot, we chose to treat monocytes that were CD14-beads-isolated from PBMCs with 50 ng/ml GM-CSF for 3 days. Macrophages were pre-treated with inhibitors (30 min) and infected with high-dose Mtb H37Rv, and LDH release and IL-1 β secretion was measured at d 24 hours. These results are included in the revised version of the paper as Fig. 1e and Supplementary Fig. 1g, discussed page 6 (Results) and page 20 (Discussion).

Similar to THP1s and primary monocytes, IL-1 β release was clearly NLRP3 inflammasome dependent as VX765, MCC950, or excess KCl abolished IL-1 β release 24 hours post infection. LDH release was somewhat reduced, although only significantly so by KCl and MCC 950, perhaps due to limited efficacy of caspase inhibitors in preventing pyroptosis (Schneider et al. 2017, PMID 29281832) and the fact that blocking one cell death pathway can unleash other pathways. The activity of the NLRP3 inflammasome also varies between cell types and cell states. In a recent report from Eicke Latz' lab, even in response to the canonical NLRP3 trigger LPS and nigericin a low proportion (<5%) of human macrophages formed ASC specks, ascribed to a regulatory role of alternative NLRP3 isoforms (Hoss et al. 2019, PMID: 31324763). Accordingly, the poorer inhibition of pyroptosis we observed in macrophages than in monocytes/THP-1s is likely due to a lower frequency of macrophages responding with inflammasome activation and IL-1 β release, which are the cells prone to undergo pyroptosis. This is added to the Discussion page 20.

Also, as pointed out by the reviewer, the major novelty demonstrated is ESX-1 mediated, contact-induced plasma membrane damage, causing K⁺ efflux and activation of the NLRP3 inflammasome. We have now confirmed this finding in human monocytes and fully differentiated macrophages (also see later for ALG2-recruitment to Mtb at the plasma membrane of macrophages in new Suppl. Fig. 6a in the revised manuscript).

Although evidence from human biopsies or infected individuals indeed would be of great interest, this paper focuses mainly on the temporal occurrence of events leading to inflammasome activation and pyroptosis during Mtb infection in single cells. These kinds of experiments are necessarily restricted mainly to cell-line systems with expressed fluorescent markers, limiting our choice of experimental systems. Moreover, even in vitro dead cells are rapidly cleared by surrounding macrophages and thus only available for analysis for a short while, so it is unclear to which extent the events we have studied would be readily detectable in fixed in vivo systems. We are currently establishing human induced stem-cell approaches to explore Mtb-induced cell death pathways in different macrophage populations further. Since these are genetically editable, we hope they can shed light on some of the unresolved questions.

The second is that this is yet another instance that demonstrates an ostensibly important influence of the RD1 encoded genes on the pathogenesis of tuberculosis, yet we are no further towards a mechanism such that this might be inhibited directly, rather than the downstream consequences. Where does the translational field go with this? Selective potassium channel blockade?

We agree that many studies have shown the importance of RD1 genes for Mtb virulence, and e.g. inhibiting ESX-1 activity directly as an antibiotic treatment would naturally be of great interest clinically. We are aware of drug-screening efforts to identify ESX targeting compounds for this purpose. Regarding our experiments focusing on host cell consequences, any perturbations to immune function tend to be double-edged (e.g. containing Mtb while exacerbating pathology, or opposite) and consequences hard to predict. If we were to speculate then we believe that compounds skewing pyroptotic or other forms of necrotic cell death, which is lytic and highly inflammatory, to apoptosis-like death while retaining a moderate release of IL-1 β would be interesting to investigate further. We have expanded this point in the discussion (page 24) with the following text:

“As shown here and elsewhere^{32,33,59,103} multiple forms of programmed necrotic pathways, including pyroptosis, necroptosis and others occur during Mtb infection. Likely Mtb simultaneously exploits these multiple pathways to spread within a host. Cell necrosis pathways including pyroptosis are thus attractive as possible targets for host-directed therapeutic strategies by limiting both bacterial spread and excessive inflammation¹⁰⁴.”

Thirdly it is also clear that the determinants of apoptotic, necrotic and pyroptotic cell death modes are indeed quite liable to influence progression or resolution of an infection. All three modes are demonstrated indeed in these experiments. Why do the authors think this happens? What are the critical early cellular determinants?

Much of the controversy regarding Mtb and cell death likely stems from the fact that most types of cell death do occur during Mtb infection. Our single cell studies indicate that apoptotic pathways are likely regulated mainly by cell-external factors, as apoptosis is similar in infected and uninfected bystander cells. On the other hand, necrosis in the form of pyroptosis and necroptosis (and possibly other pathways such as ferroptosis (Amaral et al. 2019, PMID: 30787033) likely are triggered depending on the order and type of stresses a cell experiences, cell ontogeny and phenotype, which will differ temporally and spatially over the course of infection depending on the tissue environment and bacterial burden. As we show here, membrane damage caused by high doses of Mtb at early (first 24h of infection) timepoints is closely linked to pyroptosis, but due to a form of tolerance leading to a blunted pyroptosis response (e.g. through extended TLR signaling), membrane damage at later timepoints could instead give rise to other forms of necrosis through yet to be identified pathways. Thus, the number of bacteria encountered by a host cell together with the state of the cell likely determines the outcome of this encounter. In addition, the virulence of different Mtb strains will most likely also influence cell death. There is evidence that Mtb produce factors to counter

apoptosis, inflammasome activation and more, and how effective these efforts are may additionally influence the outcome. We have highlighted this point in our considerably revised discussion, in particular on page 21 we now state:

“In summary, our results highlight that Mtb-induced pyroptosis is an independent pathway distinct from other programmed necrotic pathways such as necroptosis, although both pathways and others can and likely do occur during Mtb infection *in vivo*. Which pathway is triggered first is likely determined both by the state of the host cell (e.g. activation state, immune environment or previous stimulation) and the nature of the damage by Mtb, where more severe damage causes K⁺ efflux and pyroptosis, while lesser damage could allow e.g. TNT secretion and subsequent activation of necroptosis.”

A few specific points

1. Although the figure legends are already quite long, I found it difficult to view them in isolation to work out what had been done and why. The emphasis in the legends is very technical mentioning MOI and drug doses rather than giving any hint what the bigger question under examination is.

We thank the reviewer for bringing this to our attention. We have critically revised the figure legends to shorten methodological details and highlight the purpose of the experiments. Some experimental details have been moved to materials and methods.

2. Likewise many abbreviations are introduced from the abstract onwards without being spelt out. This again was not an aid to the non-specialist.

We thank the reviewer for bringing this to our attention. We have revised the manuscript ensuring that abbreviations are spelled out in each section.

3. The experiments using silica were interesting in the context of a tuberculosis study. Silicosis is a major risk factor for tuberculosis in mining communities and yet no mention of this is made by the authors in their discussion. I thought more could potentially have been made of this.

We thank the reviewer for pointing out this interesting connection. We have added the following paragraph in the discussion to highlight this (page 24):

“In particular, silicosis is a known risk factor for tuberculosis and studies have shown both increased phagocytosis and impaired macrophage function upon co-exposure to silica and Mtb (Konečný et al., PMID: 30687311). Due to the similar progression of phagosomal damage and PM damage during Mtb and silica exposure, we hypothesize that these processes could be exacerbated during simultaneous exposure, driving pathology. Overall, our findings point to PM damage as a shared mechanism for this group of NLRP3 activators, extending the relevance beyond tuberculosis disease.”

Reviewer #2 (Remarks to the Author):

In this manuscript by Beckwith et al, the authors demonstrate that Mtb-induced damage to the host cell plasma membrane activates the inflammasome and pyroptosis, facilitating the spread of Mtb to neighboring cells. Overall, using live cell imaging and CLEM, the author's carefully study the sequence of events in which ESX-1 mediated contact induces plasma membrane damage from either extracellular or cytosolic bacilli; this membrane injury allows K⁺ efflux, which triggers inflammasome activation, IL-1 β release, and pyroptosis unless the membrane damage is balanced by repair mediated by the endosomal sorting complexes required for transport (ESCRT) machinery. This is a

well-written, interesting study with well-controlled experiments, and the findings help clarify an important aspect of Mtb biology, as well as suggesting that plasma membrane damage is a common mechanism utilized by other NLRP3 activators that have previously been shown to act through lysosomal damage. This is an interesting and important paper, and I have only minor concerns/comments.

We thank the reviewer for understanding and appreciating our paper.

1. In the discussion about Gal3 recruitment to bacteria (lines 214-229), the authors suggest that Gal3 is recruited to bacteria with no visible phagosomal membrane and that the membrane that is visualized in Figure 4d and 4e (region 2) is an autophagosome. Does this mean the authors do not think Gal3 is binding to host glycans on the luminal surface of the phagosome? Is it directly binding Mtb?

We thank the reviewer for pointing out this imprecise wording on our part. Based on previous studies of Gal-3 binding, as well as our higher resolution live cell imaging data, we do not believe Gal-3 directly binds Mtb, rather that it binds the phagosomal luminal membrane as the reviewer indicates and as previously reported. This Gal-3+ membrane then remains in the vicinity (but not immediately adjacent to) Mtb. Soon after Gal-3 recruitment to damaged phagosomes, we also see recruitment of LC3B, indicative of autophagy. We have removed any confusing notion about recruitment of Gal-3 directly to Mtb throughout the manuscript.

Moreover, we have addressed the potential binding of Gal-3 directly to Mtb using recombinant human Gal-3 followed by anti-Gal-3 Ab staining. First, we confirmed that rhGal-3 bound to the plasma membrane of intact THP1 cells, however no binding could be observed to Mtb:

Figure 1 Recombinant human Galectin-3 (1 $\mu\text{g}/\text{ml}$) was added to the media of live adherent THP1 macrophages on ice for 3 minutes, cells were washed briefly with PBS before fixed in 4% PFA 10min at RT. Anti-Galectin-3 antibody was added for 3h, before a secondary fluorescent antibody were added for 1h. The cells were imaged on a Zeiss LSM 880 microscope.

Figure 2 Recombinant human Galectin-3 (1 $\mu\text{g/ml}$), anti-Galectin-3 antibody and secondary fluorescent antibody were incubated with live *M. Tuberculosis* (H37Rv mc26206::EBFP2) bacteria in suspension for 1h, spun down and resuspended in PBS. Bacteria were added in 96-well glass bottom microplates and imaged on an EVOS 2 widefield fluorescent microscope.

2. Of the Mtb that become Gal3+, the author's indicate that only 20% are found in an acidified compartment, concluding that autophagy is inefficient. What percent become LC3+? Is it sequestration into an autophagosome that is inefficient or maturation to an autolysosome?

We thank the reviewer for this interesting comment. About 98% of Gal-3+ compartments become LC3B+ within a few minutes, which we have now clarified in the text. If it is incomplete closure or maturation deficiencies which underlie the lack of acidification, we cannot conclusively determine, but the Gal-3+ compartments we have investigated by FIB-SEM have all shown patches of cytosolic access, indicating that incomplete closure might be the main issue. We have now added this information to the paragraph on phagosomal damage, (page 13) in the revised manuscript:

"Indeed, we observed recruitment of LC3B around intracellular Mtb shortly after recruitment of Gal-3 in 98% of events (Fig. 4f, g and Supplementary Movie S10), suggesting that ruptured Mtb phagosomes are targeted by autophagy⁶⁰⁻⁶². To investigate if autophagic targeting of Mtb lead to formation of mature, acidified autophagosomes, THP-1 macrophages expressing Gal3-mScarlet were labelled with LysoView633 (Supplementary Fig. 4a). Only ~20% of Mtb in ruptured phagosomes were later found in acidified compartments. Together with the FIB-SEM results of Mtb-associated Gal3+ phagosomes showing retained cytosolic contact between the bacteria and the host cell, this indicates that autophagosomal sequestration of Mtb after phagosomal rupture is inefficient (Supplementary Fig. 4b)."

3. Lines 395-399: The author's might comment/acknowledge that although IL-1 β is central to a successful immune response in vivo, it is not NLRP3 dependent (in mice).

We thank the reviewer for raising this point, and we have now made mention of these studies in the introduction page 3, adding the following sentence:

“However, NLRP3-independent routes to IL-1 β release have been reported in mouse infection models, leaving the role for NLRP3 in vivo less clear^{17,18}”

4. The authors claim that plasma membrane damage, occurring during phagocytosis or from the cytosol after phagosomal damage, is the cause of K⁺ efflux and subsequent NLRP3 activation by Mtb. In support of this, they observed ALG-2 recruitment to Mtb in association with Gal3 recruitment and independent of Gal3 recruitment in roughly equal proportions (Figure 6a-b). In Figure 6h, is the Gal3+ ALG-2-associated Mtb actually intracellular in the FIB-SEM? The image is difficult to interpret. If Gal3 can directly bind Mtb, it could bind extracellular Mtb, which might confound some of these studies.

According to our data (Fig. 2 above) Gal-3 does not seem to bind directly to Mtb. However, we agree with the reviewer that the ALG-2 events we have imaged by CLEM are quite complex to interpret. In Fig. 6h there are two separate Mtb aggregates, shown in slice 1 and 2, and both aggregates are partially phagocytosed. We believe that simultaneous recruitment of Gal-3 and ALG-2, or Gal-3 recruitment following ALG-2 can be explained by this state of partial phagocytosis, where, depending on the precise location of the membrane damage, there can be differing ALG-2 and Gal-3 recruitment patterns. One typical example would be outside PM damage occurring during phagocytosis causing ALG-2 recruitment, and further damage to the same phagosome once it is further internalized causing Gal-3 recruitment. In the manuscript, (Fig. 6b), we distinguish these unclear cases occurring during phagocytosis from events where we observe intracellular Gal-3 recruitment around the entire Mtb bacteria occurring prior to ALG-2 recruitment indicative of PM damage subsequent to phagosomal escape. It is only such events that are classified as “ALG-2 after Gal-3 events”. This is particularly evident in the TIRF experiments, where the ALG-2 recruitment events visible in TIRF occur centrally on the basal side of the cell in contact with the coverslip, where there cannot be phagocytosis occurring. We have updated our image analysis section in the materials and methods to clearly point out how we classify these events (page 32):

“For analysis of ALG-2 recruitment events compared to Gal-3 recruitment events, only events where a complete overlap between increased Gal-3 signal and Mtb was followed by an overlap with an increased ALG-2 signal were scored as “ALG-2 after Gal-3” events.”

5. There is no call out to Figure 4e in the text.

We apologize for this omission, this is now fixed, page 13 in the revised manuscript.

6. There are no error bars in Figure S1a.

We apologize for this omission and have corrected the Figure and Figure legend with the full dataset with deviations.

7. The figure 6 legend does not explain the what is surrounded by dashed circles, intact circles, or indicated by arrows.

We thank the reviewer for pointing this out, and have corrected the Figure legends accordingly.

Reviewer #3 (Remarks to the Author):

In this manuscript “Plasma membrane damage causes NLRP3 activation and pyroptosis during Mycobacterium tuberculosis infection” Beckwith and colleagues investigated the activation of the NLRP3 inflammasome and role of pyroptosis in the death of human macrophages following in vitro

infection with Mtb. They showed that cells containing ASC speck formation as a result of NLRP3 inflammasome activation, undergo pyroptosis since cell death was diminished when cells were deficient in NLRP3 or gasdermin D or treated with caspase inhibitors. They also describe experiments showing that Mtb induces plasma membrane damage leading to increased calcium influx and consequently ASC speck formation. The authors conclude that Mtb triggers pyroptosis through a mechanism associated with plasma membrane perturbation occurring prior to and/or after phagocytosis and that this is the major mechanism of cell death in Mtb infected human macrophages.

Overall, the manuscript is clearly written and the authors have effectively and elegantly employed different strategies of single cell imaging analysis to support their claims. A major concern with this manuscript is that the authors performed most experiments using PMA treated THP-1 cells which are commonly used as a model of human macrophages for the study of inflammasome activation because they express high levels of NLRP3, ASC and pro-caspase-1. It is well known that PMA stimulates a time- and concentration-dependent rise in cytosolic calcium levels. In addition, PMA has been reported to induce pro-IL1b, pro-caspase-1 and NLRP3 expression and, under some conditions triggers NLRP3 inflammasome activation. Also, PMA has been shown to increase the expression of P2X7R which is important for both NLRP3 inflammasome activation as well as necrotic cell death induction by facilitating calcium influx. Although these responses will not trigger pyroptosis by themselves, they may bias the induction of a pyroptotic pathway of cellular necrosis following Mtb infection. In the absence of more extensive experiments with non-PMA stimulated primary macrophages, the authors need to seriously address these concerns/provisos in the Discussion of the manuscript

We thank the reviewer for pointing this out. We are fully aware of the limitations in using cell lines, but the expression of fluorescently tagged proteins is a prerequisite to study spatiotemporal events during live Mtb infection and this cannot be done in primary cells. However, as reported in response to reviewer #1 above we have now further expanded our investigations into primary human macrophages and highlighted similarities and differences to THP-1 cells both in the results section (Fig. 1 and page 6) as well as in our considerably revised discussion (pages 19-20). We have also pointed out the particular challenges of PMA-differentiated THP-1 cells in the discussion (page 19-20):

“In addition, there are cell type differences in NLRP3 activity. It has previously been reported that monocytes require less stimulation than macrophages to activate an NLRP3 mediated inflammatory response⁸⁷, and PMA-differentiation of THP-1 cells necessary for establishing an adherent, macrophage-like phenotype also upregulates pro-inflammatory cytokines and NLRP3 pathway components thus potentially making them more prone to activation of this pathway⁸⁸.”

In one experiment (Figure 1D) the authors did employ human monocyte derived macrophages to support their claims of MTb induced pyroptosis induction based on their studies with PMA stimulated THP-1 cells. They observed a reduction in LDH release under conditions of caspase1 and NLRP3 inhibition. Nevertheless, this decrease in cell death was minor (roughly 10-40% of the normalized value of LDH release) leaving unaddressed the mechanism(s) responsible for the remainder of the death observed.

We agree with the reviewer that THP-1 cells are prone to inflammasome-activation and pyroptotic cell death as discussed above. However, we clearly show in Fig. 1f that there are multiple types of cell death occurring partly simultaneously, so partial reductions in cell death upon inhibition of one pathway is expected.

As discussed in response to concerns by reviewer #1, we have also expanded our investigations to include primary human macrophages. Results are included in new Fig. 1e (and ALG-2 recruitment in new Supplementary Fig. 6a) of the revised manuscript (see response to Reviewer 1). In brief, similar to THP1s and primary monocytes, IL-1 β release was clearly NLRP3 inflammasome dependent as VX765, MCC950, or excess KCl abolished IL-1 β release 24 hours post infection. LDH release was also reduced in response to KCl and MCC950 (NLRP3 inflammasome inhibition) but not significantly in response to caspase 1 inhibition (VX765). In addition to the new results, we have also expanded our discussion on the causes of partial responses to inhibitors, and the following text is now included (page 20):

^{48,49}“Caspase inhibitors have differing efficacies in preventing pyroptosis despite potent inhibition of IL-1 β release⁸², and there is also cross-talk between cell death pathways and caspase redundancy during inflammasome activation^{83–86}. Together with long time courses of infection, these factors could contribute to the smaller effect of caspase-1 inhibition compared to NLRP3 inhibition on cell death in primary cells, which is consistent with previous reports of caspase-independent cell death during Mtb infection^{48,49}.”

Importantly, the poorer inhibition of pyroptosis we observed in macrophages than monocytes/THP-1s is likely due to a lower frequency of macrophages responding with inflammasome activation and IL-1 β release, which are the cells prone to undergo pyroptosis (Hoss et al. 2019, PMID: 31324763). We have also included further data and discussion on the possible role of necroptotic cell death as an alternative cell death pathway. Previous data on THP-1 cells are found in Supplementary Fig. 1g, and we have now included results on the effect of RIPK3 inhibition on cell death in human monocytes and macrophages in Supplementary Fig. 1h-i. In summary, we are not able to inhibit cell death through necroptosis inhibitors in THP-1 cells but do find a role for necroptosis in primary monocytes as an additional cell death pathway. These results are now discussed more thoroughly in the discussion (page 21).

Moreover, in contrast to the experiments involving THP-1 cells, the cell death of these human monocyte derived macrophages measured by LDH release was not confirmed by DRAQ7 dye staining.

DRAQ7 staining of pyroptotic monocytes and macrophages is challenging as their nuclei tend to detach over longer time-courses to a larger extent than THP1 cells, prohibiting quantification by microscopy. However, we have verified that LDH release and DRAQ7 staining gives similar results in THP1 cells and MDMs and have included these results in Supplementary Fig. 1c, g in the revised manuscript.

Another confusing finding concerns the role of Gasdermin D which in contrast to caspase 1 or pan caspase or NLRP3 inhibition appears to account for all of the cell death observed in ThP1 cells. The authors fail to comment on this discrepancy and whether or not it may suggest the involvement of other death pathways. While the authors do acknowledge the possible participation of other death mechanism in their cultures (e.g. figure 1F) these are not adequately investigated considering their major role based on the relatively minor effects of the inflammasome inhibitors studied.

We thank the reviewer for raising this issue. The graph of DRAQ7 influx in Fig. 1e (Fig. 1c in the revised manuscript) only investigates the first 20 minutes after ASC-speck formation, showing that ASC speck formation drives rapid GSDMD-dependent pyroptosis during Mtb infection. However, GSDMD deficient cells still die after ASC-speck formation, although after a delay of one to several hours (an example of this can be seen in Supplementary Movie S6), a finding in line with recent literature showing that other targets of caspase-1 and other enzymes can trigger alternative forms of cell death in the absence of GSDMD (Schneider et al. 2017, PMID 29281832; Tsuchiya et al. 2019,

PMID: 31064994; Davis et al. 2019, PMID: 30796192). We have now clarified the interpretation (page 6).

Indeed, as pointed out above and discussed and shown in Fig. 1, other forms of cell death do play a role in Mtb infection. As mentioned above we have expanded the results to better address the contribution of necroptosis. However, the purpose of this study was to investigate the causes and consequences of inflammasome activation in single cells during Mtb infection using live-cell microscopy. In THP1 cells, blocking inflammasome activation seems to inhibit/delay cell death in the corresponding fraction of cells where inflammasome activation is induced (ASC specks). Due to the lack of specific reporters we cannot study the other necrotic pathways such as necroptosis or ferroptosis in a similar way as pyroptosis, and thus a more detailed study of these pathways are not within the scope of our study.

The authors argue that plasma membrane damage is induced both as a consequence of contact with extracellular bacteria and as result of Mtb phagocytosis. This conclusion is based on cell treatment with the phagocytosis inhibitor (cytD). Indeed, ASC speck formation was partially reduced (by 10% it appears) when cells were treated with cytD at 5uM. However, they did not independently measure the effect of the drug on Mtb uptake or test higher concentrations of CytD that have been documented previously to be more effective in preventing Mtb phagocytosis. For this reason, it is difficult to assign a role for extracellular Mtb in triggering membrane damage and no further experiments were presented to examine such a mechanism/pathway.

We apologize for omitting the data on the effect of CytD on Mtb uptake. We have now included these data in Supplementary Fig. 7a. From these data it can be seen that at 5µM CytoD phagocytosis of Mtb is strongly inhibited, so we believe our reasoning about this is consistent. Higher concentrations of CytD were toxic to the cells in our assays. The second line of reasoning is that we see ALG-2 recruitment near Mtb without prior Gal-3 recruitment in roughly half of the total ALG-2 events (Fig. 6b). Taken together, our interpretation of these events is that some of the plasma membrane damage events leading to NLRP3 inflammasome activation are caused by extracellular Mtb.

Minor point:

In most assays performed the number of cells analyzed (n=27-82, depending on the experiment) is unusually low (less than 0.1% of the total number of cells in the culture) and may not offer a fair representation. This to me is an inherent weakness in the authors' data.

For the assays at fixed timepoints we image several thousand cells per well, a substantial portion of the total cell number. However, to resolve the events we wish to investigate with sufficient spatial and temporal resolution we have to limit the number of cells imaged, as is usual in all live-cell imaging studies. To mitigate this trade-off, we perform several pilot experiments and independent experimental repeats, image random fields of view from the well, and analyze all cells present within the field of view. We believe these approaches limit any potential bias that would be present due to imaging a smaller set of the cells. Further, the "n" for the experiments mentioned by the reviewer are the cells with events (typically ASC-speck formation or Alg-2 or Gal-3 accumulation), not the total number of cells imaged. We have more clearly stated this approach and the total number of imaged cells in the Live/fixed cell imaging section in methods (page 29):

"For quantification of cell death (by DRAQ7 signal) and ASC specks at defined time points with and without inhibitors or knock-down, 16 fields of view containing n>2000 cells in total were imaged per condition in triplicate. For time-lapse imaging by confocal, typically 6 fields of view comprising >500 cells in total were imaged at 30-45 s intervals, while TIRF/WF was done on one to four fields of view

comprising 10-40 cells at 10-30s intervals. Fields of view were defined based on plate coordinates, without bias from observation of the sample, and all cells or events within every captured field of view were included in the analysis. “

Although the manuscript presents interesting findings and the focus on human macrophages is laudable, the authors evidence for a role for pyroptosis in Mtb cell death is in disagreement with numerous studies employing murine macrophages, the papers on human monocyte derived macrophages by Welin et al. and Lee et al. 2011 they reference, as well a recently published article (Pajuelo D et al. Cell Microbiol. 2019) also employing human THP1 cells that argues for a necroptotic mechanism of cell death. For this reason, it is important that the authors experimentally strengthen their claim, more carefully rule out the involvement of additional death pathways in their in vitro system and address the controversy their findings raise.

We thank the reviewer for bringing up this important issue and agree that there are conflicting results in the literature. First and foremost, we do not claim that pyroptosis is exclusive to other cell death mechanisms, rather there are multiple cell death pathways occurring during Mtb infection. As discussed above we have expanded our results in primary monocytes and macrophages and show that here, in contrast to THP-1 cells in our studies, there is also a role for necroptosis (Supplementary Fig. 1h-i). Varying MOI, Mtb culture conditions and experimental time-points also make comparisons more challenging as they could skew the form of cell death observed. In particular, we see in our data (Fig. 1f) that pyroptosis is an early event, while necroptosis in the paper by Pajuelo et al. is analyzed after 48 hours. Studies in primary human cells have previously been hampered by the poor efficacy of the caspase-1 inhibitor YVAD-FMK in inhibiting GSDMD cleavage and pyroptosis (Schneider et al. 2017, PMID 29281832). The inhibitor for caspase-1 we used in this work – VX765 – has a higher potency in preventing GSDMD cleavage by caspase-1 (Schneider et al. 2017, PMID 29281832), which is why we might see an effect in primary monocytes, but even with this inhibitor we do not significantly reduce cell death in primary macrophages, consistent with e.g. (Welin et al. 2011). At least in mice, caspase-8 can be activated at late timepoints (several hours in this context) by the inflammasome if caspase-1 proteolytic activity is absent (Schneider et al. 2017, PMID 29281832), and has been shown to cleave GSDMD (Orning et al. 2018, PMID: 30361383). These reports highlight the point we made above that inhibition or depletion of one component late in a cell death pathway might induce compensating effects, and the total effect on cell death is not as anticipated, especially for the long and heterogeneous course of an Mtb infection. Inhibiting pathway initiators further upstream appears more potent, and previous studies in THP-1 and primary human macrophages using siRNA or inhibitors against NLRP3 support our findings for a role of NLRP3 in Mtb-induced necrosis in some systems (Wong and Jacobs 2011, PMID: 21740493; Lerner et al. 2017, PMID: 28242744). We have addressed these controversies and interpretations in a more thorough and considerably revised Discussion on page 19-21 in the revised manuscript.

REVIEWERS' COMMENTS:

Reviewer #1 (Remarks to the Author):

The authors have responded to points made and indeed performed additional experiments on variously differentiated macrophages. Whilst the results of inhibition in those experiments are not dramatic the authors are candid in their interpretation and discussion.

Reviewer #2 (Remarks to the Author):

The reviewers adequately addressed my minor concerns.

Reviewer #3 (Remarks to the Author):

The authors have systematically and thoroughly addressed my criticisms and as far as I am aware the major concerns of the other reviewers. Importantly, they have frankly acknowledged the built in limitations of their in vitro experimental system and the problems in establishing generalized physiological relevance from the results obtained. Regardless, the work performed is of excellent quality and documents a novel mechanism which although unlikely on its own to be the major pathway of cell death in Mtb infected macrophages could contribute to the overall process.